# DISCRETE DIFFUSION FOR BUNDLE CONSTRUCTION

**Teng Tu**[1†], **Ai Li**[2†], **Yunshan Ma**[3*], **Shuo Xu**[4]
**Xiaohao Liu**[1], **Haokai Ma**[1], **Liang Pang**[2], **Tat-Seng Chua**[1]
[1] National University of Singapore,
[2] Key Laboratory of AI Safety, Institute of Computing Technology, CAS,
[3] Singapore Management University, [4] Shandong University
[†]Equal contribution. [*]Corresponding author.

## ABSTRACT

As a central task in product bundling, bundle construction aims to select a subset of items from large item catalogs to build an entire bundle or, more practically, complete a partial bundle. Existing methods often rely on the sequential construction paradigm that predicts items one at a time, nevertheless, this paradigm is fundamentally unsuitable for the essentially unordered bundles. In contrast, non-sequential methods model a bundle as a set, but still face two dimensionality curses: the combinatorial space grows exponentially with both bundle length and catalog size. Accordingly, we identify two technical challenges: 1) how to effectively and efficiently model the higher-order intra-bundle relations with the growth of bundle length; and 2) how to learn item representations that remain discriminative while avoiding search directly over a huge item catalog.

To address these challenges, we propose **DDBC**, a **D**iscrete **D**iffusion model for **B**undle **C**onstruction. DDBC leverages a masked denoising diffusion process to build bundles non-sequentially, capturing joint dependencies among items without relying on a fixed decoding order, thereby partially alleviating the combinatorial challenge introduced by increasing bundle length. To mitigate the curse of large catalog size, we integrate residual vector quantization (RVQ), which compresses item embeddings into discrete codes drawn from a globally shared codebook, enabling more efficient search while retaining semantic granularity. We evaluate our method on real-world bundle construction datasets of music playlist continuation and fashion outfit completion, and the experimental results show that DDBC achieves more than 100% relative performance improvements over state-of-the-art baselines on long-bundle datasets, with competitive performance on short bundles. Ablation and model studies further confirm the effectiveness of both the diffusion backbone and the RVQ tokenizer, with gains becoming more pronounced for longer bundles and larger catalogs. Our code is available at `https://github.com/LiAi16/DDBC`.

## 1 INTRODUCTION

Product bundling has been a pervasive business strategy, which originates from conventional retailing, evolves to e-commerce, and is further adopted by generic online services, such as music and video streaming (Chang et al., 2020). A product bundle is a set of relevant items assembled to satisfy users' needs (*e.g.,* games, outfits, playlists, meal kits) (Sun et al., 2024a) and promote sales regarding sellers' pursuit. Bundle construction, *i.e.,* selecting a subset of items from the large item catalog to build an entire bundle or complete a partial bundle, is the first and foremost problem among various bundle-centric studies, such as personalized bundle recommendation (Ma et al., 2022).

Existing studies, either specifically designed for bundle construction (Han et al., 2017; Bai et al., 2019; Gong et al., 2019; Deng et al., 2021) or general sequential recommendation (Kang & McAuley, 2018; Sun et al., 2019), have an important yet often overlooked flaw: most of them are based on a sequential construction paradigm, *i.e.,* predict the next item only rather than all the items in the entire bundle, however, such a sequential construction paradigm is essentially not suitable for bundle construction. Intuitively, a bundle is not a sequence of user's interacted items, and a

user does not necessarily follow a certain sequential order to consume the items within a bundle [1]. Thereby, sequential dependencies barely exist between consecutive items in a bundle, and sequential models bring marginal benefits to bundle construction. Delving deep into the technical foundations, consider $N$ as the total number of items (item catalog size) and $k$ as the bundle length (number of items within a bundle), the theoretical space of modeling a bundle as a sequence is the permutation, *i.e.,* $P(N, k)$, while modeling it as a set by relaxing the sequential constraint will significantly downgrade the space to the combination, *i.e.,* $C(N, k)$. Nonetheless, simply discarding the sequential construction paradigm, *e.g.,* existing non-sequential construction methods (Tomasi et al., 2025; Yang et al., 2024), only partially addresses the problem, since the space of the combination is still exponential to item catalog size $N$ and bundle length $k$, which we call the two dimensionality curses.

These two dimensionality curses induce two technical challenges: First, bundle construction requires modeling the intra-bundle relations, such as similarity, compatibility, composability, *etc.* , among any possible combinations of items, *e.g.,* pair-wise, tripartite, and quaternary (Chang et al., 2020). Thus, how to effectively and efficiently preserve these higher-order relations, considering that the complexity increases exponentially with the linear growth of $k$, remains the first key challenge. Second and more seriously, the item catalogs, from which we draw items to build the bundle, are often huge. For example, $N$ could be tens of thousands or even millions on some online platforms such as Spotify or Amazon. Conventional approaches typically leverage one embedding for each item (Ma et al., 2024c), consequently, it is highly difficult to navigate through the huge candidate space and precisely pick the desired item for a certain bundle. Therefore, how to learn item embeddings that are sufficiently discriminative regarding different bundling functions while maintaining a relatively small search space poses the second technical challenge.

To tackle the above two challenges, we propose a method that leverages ***Discrete Diffusion for Bundle Construction***, named as **DDBC**. Specifically, to model the higher-order intra-bundle item relations, we introduce diffusion model as the backbone to replace the previous sequential or non-sequential solutions. Basically, the diffusion model follows a non-sequential construction paradigm, where it picks items according to the learned strategies regarding the entire bundle structure instead of following a certain pre-defined left-to-right sequential order. In terms of the second challenge caused by huge item catalog size, we leverage the residual vector quantization tokenizer (RVQ) to quantize the continuous item embedding into multiple discrete codes (Rajput et al., 2023). The codes of each item are selected from a globally shared codebook that is significantly smaller than the original item set, remarkably relaxing the dimensionality curse caused by $N$. By integrating the RVQ tokenizer into the diffusion backbone, we design our discrete diffusion model DDBC. Concretely, we treat a complete bundle $\bar{\mathbf{b}}$ as the clean state at $t{=}0$; at each forward step we randomly mask a subset of positions, eventually reaching an all-`[MASK]`. The training objective is to learn the reverse denoising dynamics $p_\theta(\mathbf{b}_{t-1} \mid \mathbf{b}_t, t)$. During inference, given a partial bundle with unknown slots marked `[MASK]`, we iteratively denoise until the bundle is fully recovered. Importantly, random masking exposes the model to rich contexts during training, thereby approximating the joint distribution modeling over bundle items and providing the flexibility to accommodate different decoding priors. Also, the item codes learned by RVQ have different levels of semantic granularity, therefore, the diffusion model can learn the bundling strategy more fine-grainedly.

Our contributions include: (1) We emphasize bundle construction should follow a non-sequential construction paradigm rather than rely on a fixed decoding order, and instantiate this view with a masked denoising process. (2) We operate the diffusion model in a vector-quantized discrete space using RVQ, which relaxes the dimensionality curse caused by huge item catalog size. (3) We provide extensive empirical evidence that our approach outperforms baselines in bundle construction, with benefits especially pronounced on larger bundles and larger item catalogs. Comprehensive ablation and model studies further verify the contribution of each component.

## 2 RELATED WORK

We review three lines of literature most relevant to our work: bundle construction, generative recommendation, and discrete diffusion models.

---

[1] Some bundles may have a sequential order by design, while here we focus on the general scenarios.

**Bundle construction** is the task of selecting a subset of items from the large item catalog to build an entire bundle or complete a partial bundle. It typically comprises two parts: (1) an encoder for users, items, and bundles, and (2) a bundle generator. On the encoder side, early advances fuse semantic features (often via multimodal encoders *e.g.,* Elizalde et al. (2023); Li et al. (2023); Radford et al. (2021); Liu et al. (2025c)) with collaborative signals (Sarwar et al., 2001; He et al., 2020) to learn stronger item embeddings (Ma et al., 2022; 2024a;c;b; Salganik et al., 2024). However, these encoders do not directly capture bundle-level structure; semantically similar items may still not co-occur, whereas real user-constructed bundles balance relevance, exhibit diversity, and maintain complementarity (Sun et al., 2024a). Most bundle generators still follow a sequential construction paradigm (Chen et al., 2019; Bai et al., 2019; Chang et al., 2021; Deng et al., 2021; Liu et al., 2025b; Han et al., 2017; Sun et al., 2024b), however, the item order within a bundle does not necessarily reflect how users construct or consume bundles. Relying on a fixed order can introduce unnecessary order bias and harm generalization by overfitting to dataset-specific sequences (Yang et al., 2024). Several non-sequential approaches have been proposed. Wei et al. (2022) predicts all bundle items in parallel with a contrastive non-autoregressive decoder, but it relies on predefined templates and fixed object types. Tomasi et al. (2025) uses a continuous-space diffusion model but only accepts text prompts as input. Yang et al. (2024) outputs an order-agnostic set in one shot; however, it lacks explicit intra-bundle relations interactions during generation. More importantly, most of these works generate bundles from scratch and do not address partial-bundle construction. Our approach targets this gap via discrete masked denoising, completing the missing items in an order-agnostic manner.

**Generative recommendation** is a paradigm that reframes recommendation as generating target item IDs rather than full-rank retrieving (Wu et al., 2024; Li et al., 2024a; He et al., 2025). The line originates from generative retrieval (Rajput et al., 2023): using residual vector quantization (Barnes et al., 1996; Lee et al., 2022), items are quantized into multiple semantic IDs from coarse to fine, and an autoregressive decoder generates these IDs conditioned on context. Subsequent work extends this paradigm to multimodal settings (Liu et al., 2024) and LLM backbones (Zheng et al., 2024; Zhai et al., 2025). As previously discussed, sequential construction paradigm is not suitable for bundling tasks; accordingly, we retain the multiple semantic-ID idea but replace the autoregressive backbone with a discrete diffusion model.

**Diffusion models** learn to generate data by inverting a forward noise process; in continuous spaces they have been widely used for images, audio, and trajectories (Ho et al., 2020; Janner et al., 2022; Kong et al., 2021; Liu et al., 2023; Yang et al., 2023a). For discrete data, diffusion extends to categorical tokens by corrupting symbols (often to an absorbing [MASK]) and denoising to reconstruct them (Austin et al., 2021; Sahoo et al., 2024). Within recommendation, diffusion has largely been applied to sequential next-item prediction, operating on item latent embedding space (Wang et al., 2023; Yang et al., 2023b; Li et al., 2024b; Liu et al., 2025a), while discrete diffusion remains comparatively underexplored (Lin et al., 2024; Ju et al., 2025). In this work, we adopt an MDLM-style discrete diffusion backbone and leverage its order-agnostic nature to better model bundles. To the best of our knowledge, we are the first to study bundle construction with discrete diffusion.

## 3 METHOD

Our framework consists of two key components: (1) RVQ to discretize item embeddings, and (2) a discrete diffusion model (DDM) that operates over the code tokens for full bundle construction. The overall architecture is illustrated in Figure 1.

### 3.1 PROBLEM FORMULATION

Let $\mathcal{I} = \{i_1, i_2, \ldots, i_N\}$ denote the item catalog and $\mathcal{B} = \{\mathbf{b}_1, \mathbf{b}_2, \ldots, \mathbf{b}_M\}$ the collection of bundles, where $N$ and $M$ denote the number of items and bundles, respectively. Each bundle $\mathbf{b} \in \mathcal{B}$ is set of items, $\mathbf{b} = \{i_{j_1}, i_{j_2}, \ldots, i_{j_{|\mathbf{b}|}}\}$, $\{j_1, \ldots, j_{|\mathbf{b}|}\} \subseteq [N]$, $|\mathbf{b}| \geq 2$. Each item $i \in \mathcal{I}$ is mapped by a feature extractor to a latent vector $E(i)$, where the feature often encapsulates semantic signals and collaborative-filtering signals depending on the datasets. We formulate the bundle construction task as: for a bundle $\bar{\mathbf{b}}$, given a partial bundle $\mathbf{b}_x \subseteq \bar{\mathbf{b}}$, predict the rest part of the bundle (the complementary item set) $\mathbf{b}_y = \bar{\mathbf{b}} \setminus \mathbf{b}_x$. The same formulation also supports from-scratch construction by setting $\mathbf{b}_x = \emptyset$.

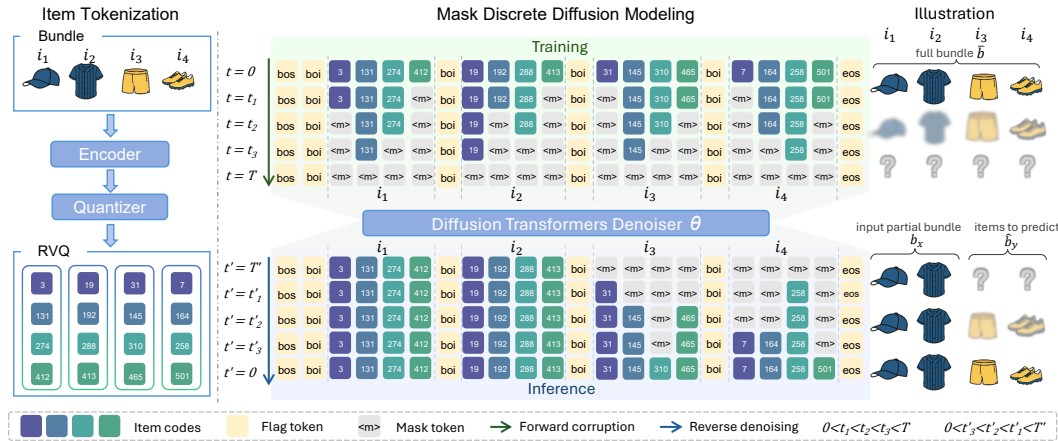

Figure 1: The overall framework of DDBC. The left side illustrates the item tokenization process via RVQ. The right side visualizes the training and inference stages of the masked discrete diffusion modeling. To be noted, we only show the forward process of the training stage, of which the backward process is just a reverse of the forward and omitted for simplicity.

## 3.2 RESIDUAL QUANTIZATION OF ITEM EMBEDDINGS

To make masked discrete diffusion feasible on large catalogs, we first discretize continuous item embeddings into a compact, hierarchical code space via RVQ. We apply an $L$-level RVQ to obtain a tuple of discrete code indices $\mathbf{z}(i) = (z_{i,1}, \ldots, z_{i,L})$. The last level is a dedup code that carries no semantics and acts purely as an auto-increment field to ensure a one-to-one mapping from a code tuple back to a unique item ID. The level order is fixed rather than permutation-equivalent: each level is learned on a different residual stage and therefore plays a distinct role in the code tuple. Formally, $z_{i,\ell} \in \mathcal{V}^{(\ell)}$ is a discrete index token for the $\ell$-th codebook, where $\mathcal{V}^{(\ell)} \triangleq \{1, \ldots, C_\ell\}$ and $C_\ell = |\mathcal{V}^{(\ell)}|$. It selects a codeword in $\mathcal{C}^{(\ell)} = \{\mathbf{e}_1^{(\ell)}, \ldots, \mathbf{e}_{C_\ell}^{(\ell)}\} \subset \mathbb{R}^d$. We view the model vocabulary as a disjoint union across levels, $\mathcal{V} \triangleq \bigsqcup_{\ell=1}^L \mathcal{V}^{(\ell)}$, so a token is always interpreted together with its level. Let the residual be $\mathbf{r}^{(0)} = E(i)$ and, for $\ell = 1, \ldots, L-1$,

$$z_{i,\ell} = \arg\min_{c \in \mathcal{V}^{(\ell)}} \left\| \mathbf{r}^{(\ell-1)} - \mathbf{e}_c^{(\ell)} \right\|_2^2, \qquad \mathbf{r}^{(\ell)} = \mathbf{r}^{(\ell-1)} - \mathbf{e}_{z_{i,\ell}}^{(\ell)}. \tag{1}$$

The reconstruction uses only the semantic codebooks: $\hat{E}(i) = \sum_{\ell=1}^{L-1} \mathbf{e}_{z_{i,\ell}}^{(\ell)}$. Early codebooks capture coarse semantics; later residual codebooks refine details, inducing semantic smoothness among similar items. We train the codebooks with an RVQ loss that combines a reconstruction term and a codebook commitment term:

$$\mathcal{L}_{\text{RVQ}} = \left\| E(i) - \hat{E}(i) \right\|_2^2 + \beta \sum_{\ell=1}^{L-1} \left( \left\| \text{sg}[\mathbf{r}^{(\ell-1)}] - \mathbf{e}_{z_{i,\ell}}^{(\ell)} \right\|_2^2 + \left\| \mathbf{r}^{(\ell-1)} - \text{sg}[\mathbf{e}_{z_{i,\ell}}^{(\ell)}] \right\|_2^2 \right), \tag{2}$$

where $\text{sg}[\cdot]$ denotes stop-gradient and $\beta$ balances the commitment loss. We use a straight-through estimator for the discrete assignment; codebooks are updated via gradient descent.

For a bundle $\mathbf{b} = \{i_1, \ldots, i_{|\mathbf{b}|}\}$, RVQ yields a token matrix $\mathbf{Z}^{(0)} \in \mathbb{N}^{|\mathbf{b}| \times L}$ with entries $z_{j,\ell} \in \{1, \ldots, C_\ell\}$; the $j$-th row contains the $L$ codes representing item $i_j$.

Among many quantization strategies, we choose RVQ for three reasons. (1) Vocabulary compression. Its theoretical capacity is $\prod_{\ell=1}^L C_\ell$, enabling a small per-level vocabulary to index a very large item universe. (2) Denser supervision. Each item contributes $L$ code tokens; at level $\ell$, a code typically aggregates roughly $N/C_\ell$ items (for $N$ total items), so code-level supervision is markedly denser than item-ID supervision. We quantify the increase in effective supervision in Section 4. (3) Coarse-to-fine granularity of semantics. Hierarchical residual codebooks perform implicit clustering at multiple granularities, which benefits downstream denoising. Additionally, when a candidate set is available in real applications, the finest semantic level and the disambiguation index may be

unnecessary at inference time: one can decode with fewer RVQ levels and map prefixes to candidate items via an inverted index over code prefixes. We study this design choice in Section 4.

## 3.3 MASKED DISCRETE DIFFUSION OVER CODE TOKENS

We cast bundle construction as a masked discrete denoising process. In contrast to the sequential construction paradigm, this formulation avoids committing to a fixed decoding order and instead lets the model condition on the full bundle context. Importantly, our discrete diffusion does not impose a strictly set-invariant objective: items are still flattened into a list and we do not explicitly enforce permutation invariance. Accordingly, the property should be understood as approximate order-insensitivity rather than exact permutation invariance. In practice, symmetric masking and aggregated supervision expose the model to diverse reveal orders, which makes the learned predictions substantially less dependent on absolute positions. To realize this idea, we adopt a discrete diffusion framework with an absorbing-mask corruption mechanism. The key components are: an input tokenization design for bundles, a forward corruption process that incrementally masks tokens, a bidirectional Transformer as the reverse denoiser, and an order-agnostic inference procedure with token-level validity constraints.

**Input tokenization for bundles.** We serialize a bundle by inserting `<boi>` before each item and wrapping with `<bos>` and `<eos>`. Let $z_{j,\ell}$ denote the $\ell$-th code token of item $j$ ($j=1,\ldots,|\mathbf{b}|$, $\ell=1,\ldots,L$). We use exactly two index sets:

$$\Omega_{\text{flag}} = \{\texttt{<bos>}, \texttt{<boi>}, \texttt{<eos>}\}, \qquad \Omega_{\text{code}} = \{(j,\ell) : j = 1,\ldots,|\mathbf{b}|, \ell = 1,\ldots,L\}. \quad (3)$$

Tokens in $\Omega_{\text{flag}}$ are never masked; corruption and prediction operate only on $z_{j,\ell}$ with $(j,\ell) \in \Omega_{\text{code}}$.

**Forward corruption.** We use an absorbing-mask Markov chain as in masked discrete diffusion. At each step $t \in \{1,\ldots,T\}$, each currently unmasked token $z_{j,\ell}$ with $(j,\ell) \in \Omega_{\text{code}}$ is independently replaced by $\texttt{[MASK]}$ with probability $\beta_t \in (0,1)$:

$$q\Big(z_{j,\ell}^{(t)} = u \mid z_{j,\ell}^{(t-1)} = u\Big) = 1 - \beta_t, \quad q\Big(z_{j,\ell}^{(t)} = \texttt{[MASK]} \mid z_{j,\ell}^{(t-1)} \neq \texttt{[MASK]}\Big) = \beta_t, \quad (4)$$

for any token value $u \neq \texttt{[MASK]}$, with the absorbing condition $q(z_{j,\ell}^{(t)} = \texttt{[MASK]} \mid z_{j,\ell}^{(t-1)} = \texttt{[MASK]}) = 1$ and independence across $(j,\ell)$. Let $\alpha_t = \prod_{s=1}^{t}(1 - \beta_s)$ be the survival probability. The closed-form transition from $t=0$ to $t$ is:

$$q\Big(z_{j,\ell}^{(t)} = v \mid z_{j,\ell}^{(0)} = u\Big) = \alpha_t \, \mathbf{1}[v = u] \; + \; (1 - \alpha_t) \, \mathbf{1}[v = \texttt{[MASK]}], \quad (5)$$

*i.e.,* after $t$ steps a token either survives with probability $\alpha_t$ or is masked with probability $1 - \alpha_t$.

**Reverse denoising.** A bidirectional Transformer $\theta$ is trained to predict the original token values from a corrupted sequence. At inference, it produces a categorical distribution for each masked position conditioned on the current noisy tokens, the timestep $t$:

$$p_\theta\big(z_{j,\ell}^{(0)} \mid \mathbf{Z}^{(t)}, t\big) \in \Delta^{|\mathcal{V}^{(\ell)}|-1}, \quad \text{where} \ \Delta^{|\mathcal{V}^{(\ell)}|-1} \triangleq \{\, p \in [0,1]^{|\mathcal{V}^{(\ell)}|} \ : \ \mathbf{1}^\top p = 1 \,\}. \quad (6)$$

Following common practice in Sahoo et al. (2024), we use the "simple" reconstruction objective that trains $p_\theta$ to predict the original token $z_{j,\ell}^{(0)}$ directly from a state $\mathbf{Z}^{(t)}$ corrupted at a random timestep $t$. Crucially, tokens that are unmasked, either because they belong to $\mathbf{b}_x$ or because they have already been generated in a previous step, are treated as clamped observations; they are never masked again and remain fixed in all subsequent steps. This mechanism enables the model to unmask items in any order during generation, without ever overwriting a code once it's decided. [2]

**Training objective.** Let $\mathcal{M}_t \subseteq \Omega_{\text{code}}$ be the set of positions masked by the forward process at step $t$. The discrete diffusion variational objective reduces to a weighted masked-token cross-entropy:

$$\mathcal{L}_{\text{NELBO}} = \mathbb{E}_{t \sim \mathcal{U}\{1,\ldots,T\}} \, \mathbb{E}_{\mathcal{M}_t} \sum_{(j,\ell) \in \mathcal{M}_t} - \log p_\theta\big(z_{j,\ell}^{(0)} \mid \mathbf{Z}^{(t)}, t\big). \quad (7)$$

---

[2] While diffusion can be extended to allow re-masking to revise earlier decisions, we do not enable that option here and leave it to future work.

**Inference.** We model bundle construction as iterative denoising of a partially masked token matrix. Let the observed set be $\mathbf{b}_x$ and the unknown complementary set be $\mathbf{b}_y$ with $|\mathbf{b}_y|$ items. Denote by $\Omega_x$ the positions (including all RVQ levels) that belong to items in $\mathbf{b}_x$, by $\Omega_y$ the positions that belong to items in $\mathbf{b}_y$, and by $\Omega_{\text{flag}}$. We construct an input sequence by flattening tokens row-wise and inserting `<boi>` before each item's $L$ tokens to mark boundaries. Let $u$ index positions in the flattened token sequence. Formally, the initial state $\mathbf{Z}$ is:

$$z_u = \mathbf{1}[u \in \Omega_x \cup \Omega_{\text{flag}}] \, x_u \;+\; \mathbf{1}[u \in \Omega_y] \, [\texttt{MASK}]. \tag{8}$$

Here, $x_u$ represents the given token at position $u$, with $\Omega_{\text{flag}}$ kept unmasked so the model knows item segmentation. The tokens of $\mathbf{b}_x$ remain clamped throughout. After decoding, a predicted item $\hat{i}_j$ is obtained by mapping its token tuple back to the catalog or to a reconstruction $\hat{E}(i) = \sum_{\ell=1}^{L-1} \mathbf{e}_{z_{i,\ell}}^{(\ell)}$.

**Token-validation constraints.** At generation time, we constrain the model's predictions to respect the level-wise token sets induced by RVQ codebooks. Specifically, for each position $(j, \ell)$ (code level $\ell$ of item $j$), we restrict the predicted token to the level-$\ell$ token set

$$z_{j,\ell} \in \mathcal{V}^{(\ell)} \triangleq \{1, \ldots, C_\ell\},$$

by setting the logits of any token not in $\mathcal{V}^{(\ell)}$ to $-\infty$ before the softmax. This validation step helps ensure that generated code tuples can be decoded to legal catalog items, which is crucial for maintaining recommendation feasibility.

## 4 EXPERIMENT

### 4.1 EXPERIMENTAL SETTINGS.

**Model settings.** We use CLHE (Ma et al., 2024c) as the shared item encoder, i.e., $E(i) = \text{CLHE}(i)$, so that performance differences can be attributed mainly to the generative backbone rather than to encoder quality. Unless otherwise noted, our RVQ uses $L = 4$ levels with fixed per-level codebook size $C_\ell \equiv C$ ($C = 128$). We utilize a lightweight DDiT architecture for our Diffusion backbone, with 6 transformer blocks, each with a hidden size of 64 and 8 self-attention heads. The model operates with a linear noise scheduler $\alpha_t = 1 - t/T$. All experiments are performed on four NVIDIA A40 GPUs, and all models are trained in 20,000 steps.

**Datasets.** Following prior research on bundle construction Ma et al. (2024c); Liu et al. (2025b), we evaluate on two representative datasets, Spotify (Chen et al., 2018) and POG (Chen et al., 2019). Unlike these works, our discrete diffusion model currently requires a fixed number of tokens per instance, so we truncate bundles to a target length. For bundles longer than the target length $k$, we sample a start index $s$ uniformly from $\{1, \ldots, |\mathbf{b}| - k + 1\}$ and keep the contiguous subsequence $(i_s, \ldots, i_{s+k-1})$. This avoids deterministic prefix truncation and reduces bias toward a fixed positional pattern. For the Spotify playlist dataset, we create three subsets by capping playlist length at 30/60/90 items (Spotify$_{k=30,60,90}$). For the POG fashion dataset, whose average bundle length is small, we start from its denser variant and derive a fixed-length version with four items (denoted POG$_{k=4}$). Unless noted, the input-predict ratio of the bundle, $|\mathbf{b}_x| : |\mathbf{b}_y|$, is set as $1 : 1$, see Table 2 for other settings. Samples shorter than the target length are dropped. Each dataset is split into train/validation/test with non-overlapping bundles. We also perform data augmentation by swapping items within the bundle, and the details are described in Appendix B.

**Candidate size.** To ensure fair comparability, we evaluate all methods under an identical candidate set, since many competitive baselines are inherently defined over a fixed candidate pool. To standardize the candidate pool, we set a candidate ratio $\rho$ and construct a shortlist $\mathcal{S}$ of size $\rho |\mathbf{b}_y|$ by augmenting the ground-truth targets with randomly sampled non-targets: $\mathcal{S} = \mathbf{b}_y \cup \text{Random}_{(\rho-1)|\mathbf{b}_y|}(I \setminus \mathbf{b})$. Unless otherwise stated, we fix $\rho = 100$ in all experiments.

### 4.2 BASELINES.

We consider both non-sequential and sequential construction methods as baselines. To isolate the effect of the generative mechanism, all the baselines use the same item features, *i.e.,* pre-trained embeddings via CLHE (Ma et al., 2024c).

Table 1: Overall performance comparison between our DDBC and baselines. "%Improv." denotes the relative improvement over the strongest baseline. Best in **bold**, second best underlined.

| Model | Spotify$_{k=30}$ | | | Spotify$_{k=60}$ | | | Spotify$_{k=90}$ | | | POG$_{k=4}$ | | |
|---|---|---|---|---|---|---|---|---|---|---|---|---|
| | F1 ↑ | Jacc ↑ | OAS ↑ | F1 ↑ | Jacc ↑ | OAS ↑ | F1 ↑ | Jacc ↑ | OAS ↑ | F1 ↑ | Jacc ↑ | OAS ↑ |
| CLHE | 0.071 | 0.039 | 0.373 | 0.100 | 0.054 | 0.446 | 0.119 | 0.065 | 0.486 | 0.140 | 0.096 | 0.446 |
| Bi-LSTM | 0.124 | 0.071 | 0.489 | 0.062 | 0.034 | 0.430 | 0.047 | 0.025 | 0.426 | 0.035 | 0.024 | 0.390 |
| SASRec | 0.070 | 0.043 | 0.318 | 0.089 | 0.054 | 0.310 | 0.050 | 0.029 | 0.285 | 0.169 | 0.114 | 0.468 |
| TIGER | 0.093 | 0.053 | 0.329 | 0.129 | 0.076 | 0.413 | 0.123 | 0.070 | 0.480 | **0.213** | **0.157** | **0.546** |
| BundleNAT | 0.153 | 0.090 | 0.454 | 0.101 | 0.056 | 0.438 | 0.095 | 0.052 | 0.446 | 0.145 | 0.097 | 0.462 |
| BundleMLLM | 0.046 | 0.024 | 0.296 | 0.045 | 0.024 | 0.324 | 0.052 | 0.027 | 0.355 | 0.070 | 0.047 | 0.322 |
| **DDBC** | **0.282** | **0.176** | **0.660** | **0.285** | **0.178** | **0.662** | **0.287** | **0.177** | **0.684** | 0.139 | 0.098 | 0.526 |
| *%Improv.* + | 84.3% | 95.6% | 35.0% | 120.9% | 134.2% | 48.3% | 133.3% | 152.9% | 40.7% | – | – | – |

**Non-sequential construction methods.** They input the partial bundle $\mathbf{b}_x$ and predict all the items in the complementary set at once. *CLHE* (Ma et al., 2024c): A method that leverages contrastive learning and hierarchical encoder to learn item and bundle representations. To be noted, CLHE was not originally designed to predict all the items in the complementary set, while it follows the typical top-k recommendation paradigm and evaluation protocol. We re-evaluate it against our metrics that are pertinent to entire bundle construction. *BundleNAT* (Yang et al., 2024): A non-autoregressive generator that predicts a set of items in one shot using preference/compatibility signals. It was originally used for the task of personalized bundle recommendation instead of bundle construction, we adapt it for our task by removing the user inputs.

**Sequential construction methods.** They follow an autoregressive construction strategy: initialize $\mathbf{s}_0 = \mathbf{b}_x$; for $j = 0, \ldots, |\mathbf{b}_y| - 1$, choose $\hat{i}_j = \arg\max_{i \notin \mathbf{s}_j} \pi(i \mid \mathbf{s}_j)$ and update $\mathbf{s}_{j+1} = \mathbf{s}_j \cup \{\hat{i}_j\}$ until $|\mathbf{s}_{|\mathbf{b}_y|}| = |\bar{\mathbf{b}}|$, where $\mathbf{s}_j$ is the current selected set after $j$ steps and $\pi(i \mid \mathbf{s}_j)$ is the model score (or probability) of adding item $i$ given $\mathbf{s}_j$. *Bi-LSTM* (Han et al., 2017): It uses bi-directional LSTM to model the bundle as a sequence. *SASRec* (Kang & McAuley, 2018): A Transformer-based sequential recommender trained for next-item prediction. *TIGER* (Rajput et al., 2023): It generates items as discrete semantic token sequences with an autoregressive decoder. *BundleMLLM* (Liu et al., 2025b): It finetunes a multimodal LLM for bundle construction. Its original evaluation is based on the multiple-choice question protocol since it is impossible to input all the candidate items as input due to context limitation of LLMs. Even though this setting is easier than our all-ranking setting, to be simple, we follow this paradigm and set the candidate set as 20.

To be noted, many other recommendation models can be adapted as baselines by following the paradigm of either the sequential or non-sequential construction. For example, the baselines implemented in Ma et al. (2024c): MultiDAE (Wu et al., 2016), MultiVAE (Liang et al., 2018), Hypergraph (Yu et al., 2022), and Transformer (Wei et al., 2023), *etc.* or the other advanced sequential recommendation models. However, we do not include them because they either underperform CLHE or are less relevant to the bundle-construction setting.

## 4.3 EVALUATION METRICS

We report retrieval-based metrics F1 and Jaccard (Jacc) (Manning et al., 2008; Ding et al., 2023), as well as Optimal Alignment Score (OAS) (Kuhn, 1955). Higher F1, Jacc, and OAS indicate better performance. Let $\hat{\mathbf{b}}_y$ denote the set of predicted items, these metrics are calculated by:

$$\text{F1} := \frac{2PR}{P+R}, \quad \text{Jacc} := \frac{|\hat{\mathbf{b}}_y \cap \mathbf{b}_y|}{|\hat{\mathbf{b}}_y \cup \mathbf{b}_y|}, \quad \text{OAS} := \frac{1}{|\mathbf{b}_y|} \max_{P^*} \sum_{(i,j) \in P^*} \cos\big(E(i), E(j)\big) \quad (9)$$

where $P = \frac{|\hat{\mathbf{b}}_y \cap \mathbf{b}_y|}{|\hat{\mathbf{b}}_y|}$, $R = \frac{|\hat{\mathbf{b}}_y \cap \mathbf{b}_y|}{|\mathbf{b}_y|}$, and $P^*$ is the optimal matching between items in $\hat{\mathbf{b}}_y$ and $\mathbf{b}_y$, and $\cos(\cdot, \cdot)$ denotes cosine similarity. Previous methods in bundle construction use popular next-item recommendation metrics, such as recall, ndcg, or hit rate (Ma et al., 2024c). However, these metrics are not suitable in the scenario of full bundle construction, which needs to assess the quality of the predicted entire item set instead of single item. Therefore, we adopt these three complementary set-level metrics to measure bundle-construction quality more faithfully than next-item recommendation metrics in our setting.

Table 2: Effect of input-predict ratio on Spotify$_{k=60}$. Best in **bold**, second best underlined.

| Model | 5/55 | | | 10/50 | | | 30/30 | | | 45/15 | | |
|---|---|---|---|---|---|---|---|---|---|---|---|---|
| | F1 ↑ | Jacc ↑ | OAS ↑ | F1 ↑ | Jacc ↑ | OAS ↑ | F1 ↑ | Jacc ↑ | OAS ↑ | F1 ↑ | Jacc ↑ | OAS ↑ |
| BundleNAT | 0.106 | 0.059 | 0.443 | 0.128 | 0.072 | 0.463 | 0.101 | 0.056 | 0.438 | 0.084 | 0.046 | 0.359 |
| SASRec | 0.119 | 0.070 | 0.425 | 0.131 | 0.078 | 0.442 | 0.089 | 0.054 | 0.310 | 0.095 | 0.055 | 0.315 |
| TIGER | 0.087 | 0.050 | 0.365 | 0.100 | 0.059 | 0.381 | 0.129 | 0.076 | 0.413 | 0.154 | 0.091 | 0.426 |
| **DDBC** | **0.237** | **0.144** | **0.637** | **0.268** | **0.164** | **0.664** | **0.285** | **0.178** | **0.662** | **0.260** | **0.161** | **0.614** |
| *Improv. +* | 99.2% | 105.7% | 43.8% | 104.6% | 110.3% | 43.4% | 120.9% | 134.2% | 48.3% | 68.8% | 76.9% | 44.1% |

Table 3: Effect of candidate ratio on Spotify$_{k=60}$. Best in **bold**, second best underlined.

| Model | $\rho$=10 | | | $\rho$=20 | | | $\rho$=50 | | | $\rho$=100 | | |
|---|---|---|---|---|---|---|---|---|---|---|---|---|
| | F1 ↑ | Jacc ↑ | OAS ↑ | F1 ↑ | Jacc ↑ | OAS ↑ | F1 ↑ | Jacc ↑ | OAS ↑ | F1 ↑ | Jacc ↑ | OAS ↑ |
| BundleNAT | 0.266 | 0.163 | 0.508 | 0.210 | 0.124 | 0.485 | 0.153 | 0.088 | 0.464 | 0.101 | 0.056 | 0.438 |
| SASRec | 0.292 | 0.194 | 0.519 | 0.200 | 0.126 | 0.474 | 0.200 | 0.126 | 0.475 | 0.089 | 0.054 | 0.310 |
| TIGER | 0.191 | 0.151 | 0.326 | 0.107 | 0.080 | 0.295 | 0.108 | 0.081 | 0.296 | 0.129 | 0.076 | 0.413 |
| **DDBC** | **0.599** | **0.447** | **0.763** | **0.503** | **0.355** | **0.727** | **0.380** | **0.250** | **0.689** | **0.285** | **0.178** | **0.662** |
| *Improv. +* | 105.1% | 130.4% | 47.0% | 139.5% | 181.7% | 49.9% | 90.0% | 98.4% | 45.1% | 120.9% | 134.2% | 48.3% |

## 4.4 OVERALL PERFORMANCE COMPARISON

Table 1 shows the overall performance of DDBC compared with baseline methods. First, among the baselines, BundleNAT and TIGER achieve the strongest performance. These results respectively highlight two key component of our model: the non-sequential construction paradigm and the advantages of discretizing items into multiple codes. Second, on the Spotify dataset series, DDBC clearly outperforms all baselines, achieving a 152.9% improvement in Jacc on Spotify$_{k=90}$. Moreover, the performance gain of DDBC becomes more pronounced as the bundle length increases. These results demonstrate that DDBC effectively captures the higher-order intra-bundle item relations, particularly for long-sequence bundles with rich structural dependencies. Third, on POG$_{k=4}$, our model does not outperform TIGER. In fact, since we only predict two items, the task, to some extent reduces to a next-item prediction scenario, where autoregressive methods such as TIGER have a clear advantage.

## 4.5 MODEL STUDY

**Effect of input-predict ratio.** We conduct experiments with different input-predict ratio on Spotify$_{k=60}$ and report results in Table 2. We observe that DDBC outperforms all baselines across different partial bundle sizes and exhibits a relatively consistent performance, demonstrating its robustness in scenarios with limited known items. Specifically, when the known partial bundles are small (*e.g.*, 5/55, 10/50, 30/30), DDBC achieves substantial improvements over the best baseline by 106%, 110%, 143% on Jaccard, respectively. Although the performance gap narrows down as the number of input items grow, our method continues to maintain a leading position. These results highlight that DDBC is capable of generating coherent and distribution-aware bundles even when only a small subset of items is provided, validating the effectiveness of our masked denoising formulation and the discrete diffusion mechanism.

**Effect of candidate ratio.** We report the results for DDBC and the baselines under different candidate ratios in Table 3. The results indicate that while the absolute values of the evaluation metrics fluctuate as the candidate ratio ($\rho$) increases, the relative improvements of DDBC over all baselines remain consistently substantial. Interestingly, among baselines, when $\rho$ increases, TIGER starts to bypass other baselines on F1 and Jacc ($\rho$=100). This can be attributed to the fact that RVQ allows precise reconstruction of item IDs, implying the advantages of using RVQ. These findings highlight the adaptability of DDBC under varying candidate pool sizes, demonstrating its ability to maintain strong bundle representations even when the retrieval space becomes more challenging.

**Efficiency analysis.** We record the inference time and parameter size of DDBC and the baseline methods, as reported in Figure 2, where the circle radius indicates each model's overall performance. The inference time is measured on Spotify$_{k=60}$. Specifically, although Bi-LSTM has fast inference and smallest parameters, its performance is not competitive (see Table 1). DDBC is highly parameter-efficient, containing only 0.79M parameters, and is significantly smaller than other base-

Table 4: Ablation study of key components.

| Variant | F1 | Jacc | OAS |
|---|---|---|---|
| Our proposed DDBC | 0.282 | 0.176 | 0.660 |
| *w/o RVQ* | 0.021 | 0.011 | 0.557 |
| *w/o boi token* | 0.176 | 0.104 | 0.538 |
| *w/o data augmentation* | 0.254 | 0.158 | 0.599 |
| *w/o token validity filter* | 0.276 | 0.173 | – |

Table 5: Effect of RVQ levels.

| $\ell \in$ | F1 | Jacc | OAS |
|---|---|---|---|
| $\{1\}$ | 0.182 | 0.108 | 0.556 |
| $\{1, 2\}$ | 0.224 | 0.136 | 0.589 |
| $\{1, 2, 3\}$ | 0.240 | 0.148 | 0.590 |
| *Our proposed DDBC* | | | |
| $\{1, 2, 3, 4\}$ | 0.282 | 0.176 | 0.660 |

lines. Moreover, DDBC's inference speed is comparable to the one-shot generation method Bundle-NAT and faster than all other baseline models; in particular, it is substantially faster than BundleM-LLM, which relies on interactions with large language models.

## 4.6 ABLATION STUDY

**Key components.** To further evaluate the effectiveness of key components of our model, we conduct ablation experiments (Table 4) to assess the contribution of each design choice in DDBC. Considering the resource and time overhead imposed by the extremely large vocabulary in Spotify (254,155), performing ablation studies without RVQ would be prohibitively expensive. Therefore, we adopt Spotify$_{k=30}$, the shortest sequence setting, as a more practical benchmark for these experiments.

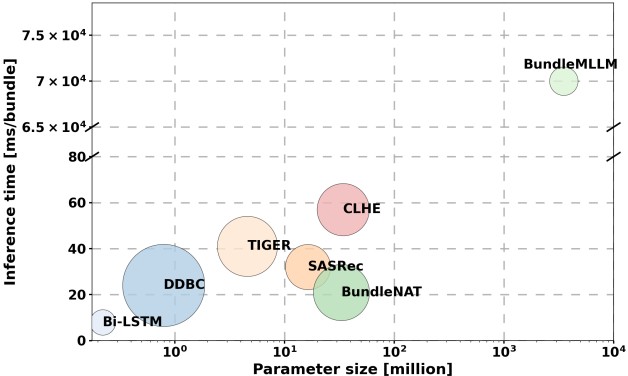

Figure 2: Illustration of the model efficiency comparison. The x-axis is parameter size (millions), the y-axis is inference time (ms/bundle), and bubble radius corresponds to overall performance (larger is better).

We study how each component contributes to performance: hierarchical (coarse-to-fine) decoding, token validity filter, boi token, data augmentation, we also investigate how different RVQ depth affect performance. We derive the following insights. (1) Removing RVQ results in a dramatic performance drop. We encode items using their IDs and initialize their embeddings with CLHE features, results in a dramatic drop in both F1 and Jaccard. This demonstrates RVQ mitigates the dimensionality curse due to $N$, which is crucial as it permits dense supervision. (2) Discarding the boi token leads to a performance decline. Since Diffusion's generation lacks inherent sequence, integrating the boi token is necessary to guide the model with positional information. (3) Data augmentation shows beneficial for modeling. The results of training on the original dataset show a slight performance reduction in this case, and simple data augmentation remains significant for diffusion modeling because it explicitly provides richer input context. (4) The token validity filter remains essential to guarantee the validity of the generated bundles, despite its removal leading to only a marginal decrease in performance. We evaluated the necessity of the token validity filter during inference. While removing the filter resulted in only a marginal decrease in overall performance, the invalid ratio concurrently rose to 2.5%. Therefore, the filter remains essential to guarantee the validity of the generated bundles.

To investigate the impact of the item embedding quantization level on model performance as discussed in method section,with a fixed 4 levels RVQ, we train DDBC with utilizing different levels of RVQ. We report the results in Table 5. As the number of RVQ levels used increases, the model captures increasingly finer-grained item information, leading to substantial improvements in all the evaluation metrics. We state that the current setting represents a favorable trade-off between the representational capacity and compression ratio of RVQ.

## 5 CONCLUSION

We recast bundle construction with a masked discrete diffusion model that progressively resolves unknown items in an order-insensitive manner. Conceptually, the formulation addresses the dual dimensionality challenges of bundle construction: (i) it removes the need to commit to a fixed decoding order, reducing the search space from permutations to combinations while better preserving higher-order item relations; and (ii) it shrinks the effective search space by mapping items to codes drawn from a globally shared codebook. Empirically, coupling DDM with RVQ yields consistent gains over prior sequential and non-sequential construction baselines, with especially strong improvements as bundle length grows.

**Discussion.** Our current instantiation is trained under a fixed-length protocol. While Appendix C.3 shows that DDBC already supports several training-free variable-length generation strategies, more principled stopping criteria and explicit length control remain open. Personalization is currently mediated by frozen encoders for user–item signals and item semantics; introducing explicit conditioning into the diffusion process (*e.g.,* context features or user instructions) could yield more user-specific bundling. The RVQ design space (*e.g.,* number of levels, codebook sizes, and training regimes) also deserves further study to balance identifiability, compression, and semantic smoothness. Evaluation also remains imperfect: our current protocol relies on a random candidate pool for comparability with candidate-conditioned baselines and for statistical stability of exact-match metrics, but this only partially reflects the open-world nature of bundle construction where multiple item substitutions may be equally valid beyond a restricted pool; developing evaluation procedures that better capture solution diversity while remaining reproducible is an important direction. Finally, diffusion schedules and inference policies merit deeper optimization: adaptive timestep schedules, selective re-masking strategies, and entropy-guided decoding may further improve sample efficiency and robustness.

## ETHICS STATEMENT

We affirm compliance with the ICLR Code of Ethics. Our study addresses bundle construction (*e.g.,* playlists and fashion outfits) and uses publicly available research datasets and splits released by prior work; no personally identifiable information (PII) or sensitive attributes are collected, inferred, or released. The inputs consist of item identifiers and non-sensitive metadata, and our models operate on discretized representations without accessing user profiles. Any code and models we release will be for research use only and will not include copyrighted media or proprietary assets.

## REPRODUCIBILITY STATEMENT

We make our method reproducible by specifying the full training and evaluation pipeline, including the RVQ configuration, diffusion horizon, architecture, schedulers, and all hyperparameters. We provide a repository `https://github.com/LiAi16/DDBC`, including the implementation of our model as well as the evaluation scripts for F1, Jaccard, and OAS. Upon publication, we plan to release checkpoints (where licenses permit) to reproduce all main and ablation results.

## ACKNOWLEDGMENTS

This research was supported by the NExT Research Center; by the Strategic Priority Research Program of the Chinese Academy of Sciences (CAS) (Grant No. XDB0680302), the National Natural Science Foundation of China (NSFC) (Grant No. 62276248), and the Beijing Nova Program (Grant No. 20250484765); and by the Singapore Ministry of Education (MOE) Academic Research Fund (AcRF) Tier 1 grant.

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

## A    THE USE OF LARGE LANGUAGE MODELS

We used large language models (*e.g.,* OpenAI ChatGPT 5, Claude Sonnet 4.5) as an assistive tool for writing polish (grammar, phrasing, and LaTeX formatting), troubleshooting LaTeX errors, and scaffolding non-critical scripts (plotting and small utilities). LLMs did not contribute novel scientific ideas, data collection, or result selection, and any code snippets suggested by an LLM were reviewed, rewritten where necessary, and validated by the authors. All technical claims, mathematical formulations, and empirical results are the authors' responsibility. LLMs are not listed as authors and have no authorship rights.

## B    IMPLEMENTATION DETAILS

**Dataset statistics.** The statistics for the datasets used in our experiments are summarized in Table 6. All Spotify series datasets share a large item catalog size (N) of 254,155, with a massive candidate space. Density $\text{Spotify}_{k=30} > \text{Spotify}_{k=60} > \text{Spotify}_{k=90}$. Conversely, the $\text{POG}_{k=4}$ dataset features a substantially smaller item catalog size (N) of 31,217 and contains a more consistent number of bundles across the splits (*e.g.,* $\text{M}_{train} = 29,704$). This variation in item catalog size and bundle count allows for a comprehensive evaluation of our method's scalability and performance under different data density conditions.

**Data augmentation.** To improve the model's robustness and prevent overfitting to a particular sequential item order, we employ a data augmentation strategy based on item swapping. Specifically, for each original item sequence, we performed a series of adjacent item swaps on a copy of the sequence. For the POG dense dataset, we set swap ratio 0.8, and for the Spotify datasets, we used swap ratio 0.4. Subsequently, we randomly sampled a fixed-length subsequence (sequence length) from the perturbed sequence, creating a new augmented training instance. This data augmentation is an enhancement to the diffusion model, fulfilling the non-sequential modeling objective while countering the potential issue of overfitting to the certain given sequential order in the bundle.

**Over-retrieval.** To standardize the evaluation of generative models which can produce multiple possible outputs, we employ an Over-Retrieval Strategy. This strategy aggregates the results from multiple generation attempts, effectively forming the union $\hat{B}_y$ used in the retrieval-based metrics. For generative sampling models, we evaluate performance under a varying number of attempts $b \in \{1, 5, 10, 20, 50\}$ (denoted as Multiple Sampling, MS). For autoregressive baselines that use beam search, we report the results using beam width $b \in \{1, 3, 5, 10, 20, 50\}$, mapping the beam width to the number of attempts ($K = b$) for a fair comparison of computational cost.

**Input tokenization for bundles (details).** Given $b = \{i_1, \ldots, i_{|b|}\}$ with item codes $\mathbf{z}(i_j) = (z_{j,1}, \ldots, z_{j,L})$, the serialized sequence is

$$\mathbf{x} = \big(\texttt{<bos>}, \underbrace{\texttt{<boi>}, z_{1,1}, \ldots, z_{1,L}}_{\text{item 1}}, \tag{10}$$

$$\underbrace{\texttt{<boi>}, z_{2,1}, \ldots, z_{2,L}}_{\text{item 2}},$$

$$\vdots$$

$$\underbrace{\texttt{<boi>}, z_{|b|,1}, \ldots, z_{|b|,L}}_{\text{item } |b|}, \texttt{<eos>}\big).$$

Its length is $U = 2 + |b|(L+1)$. Define the index map for item $j$ and level $\ell$:

$$u(j,0) = 1 + (j-1)(L+1) + 1, \qquad u(j,\ell) = u(j,0) + \ell, \ \ \ell \in \{1, \ldots, L\}. \tag{11}$$

We then specify exactly two sets:

$$\Omega_{\text{flag}} = \{1, U\} \cup \{u(j,0)\}_{j=1}^{|b|}, \qquad \Omega_{\text{code}} = \{u(j,\ell) : j = 1, \ldots, |b|, \ \ell = 1, \ldots, L\}. \tag{12}$$

By construction, $\Omega_{\text{flag}} \cap \Omega_{\text{code}} = \varnothing$ and $\Omega_{\text{flag}} \cup \Omega_{\text{code}} = [U]$. Positions in $\Omega_{\text{flag}}$ are never masked; corruption and prediction operate only on $\Omega_{\text{code}}$ (including the dedup level $\ell = L$).

Table 6: Dataset statistics. $N$ is the catalog size (total number of items in the dataset); $M_{\text{train/val/test}}$ are the number of bundles in train/val/test sets.

| Dataset | $N$ | $M_{\text{train}}$ | $M_{\text{val}}$ | $M_{\text{test}}$ |
|---|---|---|---|---|
| $\text{Spotify}_{k=30}$ | 254,155 | 321,929 | 1,374 | 2,744 |
| $\text{Spotify}_{k=60}$ | 254,155 | 253,358 | 798 | 1,582 |
| $\text{Spotify}_{k=90}$ | 254,155 | 188,618 | 463 | 969 |
| $\text{POG}_{k=4}$ | 31,217 | 29,704 | 1,303 | 2,521 |

---

**Algorithm 1** RVQ-ENCODE for an item embedding

---

**Require:** Item embedding $E(i) \in \mathbb{R}^d$; semantic codebooks $\{\mathcal{C}^{(1)}, \dots, \mathcal{C}^{(L-1)}\}$ with $\mathcal{C}^{(\ell)} = \{\mathbf{e}_c^{(\ell)}\}_{c=1}^{C_\ell} \subset \mathbb{R}^d$; dedup indexer $\text{Dedup}(i) \in \{1, \dots, C_L\}$
**Ensure:** Code indices $\mathbf{z}(i) = (z_{i,1}, \dots, z_{i,L})$ and reconstruction $\hat{E}(i)$

1: $\mathbf{r} \leftarrow E(i); \quad \hat{E} \leftarrow \mathbf{0}_d$
2: **for** $\ell = 1$ **to** $L - 1$ **do**                                         ▷ semantic levels
3:      $z_{i,\ell} \leftarrow \underset{c \in \{1, \dots, C_\ell\}}{\arg\min} \|\mathbf{r} - \mathbf{e}_c^{(\ell)}\|_2^2$                     *(tie-break: smallest index)*
4:      $\hat{E} \leftarrow \hat{E} + \mathbf{e}_{z_{i,\ell}}^{(\ell)}; \quad \mathbf{r} \leftarrow \mathbf{r} - \mathbf{e}_{z_{i,\ell}}^{(\ell)}$
5: **end for**
6: $z_{i,L} \leftarrow \text{Dedup}(i)$                                      ▷ non-semantic dedup level
7: **return** $\mathbf{z}(i), \; \hat{E}(i) = \hat{E}$

---

**RVQ encoding pseudocode.** We elucidate the pseudocode in Algorithm 1, specifying the encoding for an item embedding.

**Evaluation metric setting.** To quantify the latent-space alignment between the predicted bundle $\hat{\mathbf{b}}_y$ and the ground-truth bundle $\mathbf{b}_y$, we define the *Optimal Alignment Score* (OAS). Specifically, for each item $i \in \mathbf{b}_y$ and $\hat{i} \in \hat{\mathbf{b}}_y$, we compute a pairwise similarity score $S(i, \hat{i})$ based on their item embeddings. This induces a weighted bipartite graph, where the two node sets correspond to items in $\mathbf{b}_y$ and $\hat{\mathbf{b}}_y$, respectively, and each edge is weighted by the corresponding similarity score.

We then seek the optimal bipartite matching that maximizes the total similarity:

$$P^* = \arg\max_P \sum_{(t_i, \hat{t}_j) \in P} S(t_i, \hat{t}_j), \tag{13}$$

where $P$ is a valid matching between tracks in $p_{\text{Target}}$ and $p_{\text{Predict}}$. Based on the optimal matching $P^*$, we define OAS as

$$\text{OAS}(p_{\text{Predict}}, p_{\text{Target}}) = \frac{1}{|p_{\text{Target}}|} \sum_{(t_i, \hat{t}_j) \in P^*} S(t_i, \hat{t}_j). \tag{14}$$

Intuitively, OAS measures how well the predicted playlist aligns with the target playlist in the latent space, even when the exact track identities do not match. A higher OAS indicates better semantic alignment and better prediction quality.

**Hungarian algorithm.** We compute the optimal matching $P^*$ using the Hungarian algorithm. Since the Hungarian algorithm solves a minimum-cost assignment problem, we convert pairwise similarities into costs via $1 - S(t_i, \hat{t}_j)$, solve the corresponding assignment, and then convert the result back into a similarity-based score. The detailed procedure is shown in Algorithm 3.

## C ADDITIONAL RESULTS

**Comparison across datasets using Jaccard.** We compare the results across datasets using Jaccard with multiple attempts. As Table 7 shows, among the established baselines, BundleNAT generally achieves the best Jaccard performance across the Spotify datasets. This suggests that BundleNAT's non-autoregressive architecture is particularly effective at generating relevant set-based results compared to the sequential models. What's more, our proposed method, DDBC, consistently and significantly outperforms all baselines across every dataset and attempt level. On $\text{Spotify}_{k=30}$, the

---

**Algorithm 2** Constraint-aware order-agnostic decoding (inference)

---

**Require:** Observed item set $\mathbf{b}_x$; index maps $u(j, \ell)$ and $\text{INVIDX}(u) \to (j, \ell)$; code-domain valid sets $\{\mathcal{V}^{(\ell)}\}$;
  diffusion model $p_\theta$; horizon $T$
**Ensure:** Clean token matrix $\mathbf{Z}^{(0)}$ and completed bundle $\hat{\mathbf{b}}$
1: Initialize $\mathbf{Z}$ with tokens for $\Omega_x$ and for $\Omega_{\text{flag}}$; set $z_u \leftarrow \texttt{[MASK]}$ for all $u \in \Omega_y$
2: **while** there exists $u \in \Omega_y$ with $z_u = \texttt{[MASK]}$ **do**
3:  Choose a timestep $t \in \{1, \ldots, T\}$ *(e.g., $t = T - s + 1$ at step $s$, or by a schedule)*
4:  **for all** $u \in \Omega_y$ with $z_u = \texttt{[MASK]}$ **do**
5:    $(j, \ell) \leftarrow \text{INVIDX}(u)$
6:    $\pi \leftarrow p_\theta(\cdot \mid \mathbf{Z}^{(t)} = \mathbf{Z}, t)$        *categorical over $\{1, \ldots, C_\ell\}$ (no `[MASK]`)*
7:    **mask out invalids:** $\pi[c] \leftarrow 0$ for $c \notin \mathcal{V}^{(\ell)}$; $\pi \leftarrow \pi / \sum_c \pi[c]$
8:    $P(u) \leftarrow \pi$
9:  **end for**
10:  Select a reveal set $S \subseteq \{u \in \Omega_y : z_u = \texttt{[MASK]}\}$ *(e.g., top-$k$ by $\max P(u)$, lowest-entropy, or*
  reveal ratio $\eta$)
11:  **for all** $u \in S$ **do**
12:    **decode:** $z_u \leftarrow \arg\max P(u)$ *(or sample with temperature/top-p)*
13:    **clamp:** $z_u$ stays unmasked thereafter
14:  **end for**
15: **end while**
16: $\mathbf{Z}^{(0)} \leftarrow \mathbf{Z}$
17: **return** $\mathbf{Z}^{(0)}$, $\hat{\mathbf{b}}$ via $\hat{i}_j = \text{CODE2ITEM}(z_{j,1:L})$ for all $j$

---

DDBC model achieves a Jacc@1 of 0.164, nearly doubling the performance of the best baseline (BundleNAT at 0.090). This performance gap confirms the efficacy and advanced capability of our model, especially when generating predictions with multiple attempts.

Table 7: Comparison across datasets using Jaccard with $A \in \{1, 5, 20\}$. Best in **bold**, second best underlined.

| Model (A/beam) | Spotify$_{k=30}$ | | | Spotify$_{k=60}$ | | | Spotify$_{k=90}$ | | | POG$_{k=4}$ | | |
|---|---|---|---|---|---|---|---|---|---|---|---|---|
| | Jacc@1 | Jacc@5 | Jacc@20 | Jacc@1 | Jacc@5 | Jacc@20 | Jacc@1 | Jacc@5 | Jacc@20 | Jacc@1 | Jacc@5 | Jacc@20 |
| CLHE | .009 | .005 | .002 | .008 | .004 | .002 | .007 | .003 | .001 | .059 | .048 | .020 |
| SASRec | .043 | .023 | .009 | .054 | .030 | .013 | .029 | .013 | .005 | .114 | .066 | .024 |
| TIGER | .053 | .028 | .011 | .076 | .036 | .013 | .070 | .034 | .013 | **.157** | **.094** | **.038** |
| BundleNAT | .090 | .076 | .033 | .056 | .055 | .027 | .052 | .052 | .026 | .097 | .055 | .023 |
| **DDBC** | **.177** | **.130** | **.053** | **.185** | **.137** | **.055** | **.177** | **.132** | **.054** | .098 | .073 | .032 |

**Latent-space quality.** To better explore the latent-space quality of the items generated by our method, we report the OAS metric at $A = 50$, as shown in Table 8. For the Spotify dataset series, when the bundle length ($k$) increases from 30 to 90, the latent-space quality becomes more stable. Specifically, the average OAS increases consistently from 0.603 (Spotify$_{k=30}$) to 0.662 (Spotify$_{k=60}$) and finally to 0.685 (Spotify$_{k=90}$). Since a higher OAS indicates higher similarity (better alignment) in the latent space, this trend suggests that our method has an improved capacity to model longer bundles.

Table 8: Latent-space quality at $A = 50$ measured by OAS↑. We report $\{\min, \text{avg}, \max, \text{var}\}$ of OAS over test bundles.

| Dataset | min | avg | max | var |
|---|---|---|---|---|
| Spotify$_{k=30}$ | 0.477 | 0.603 | 0.717 | 0.003 |
| Spotify$_{k=60}$ | 0.582 | 0.662 | 0.733 | 0.001 |
| Spotify$_{k=90}$ | 0.625 | 0.685 | 0.738 | 0.001 |
| POG$_{k=4}$ | 0.246 | 0.525 | 0.791 | 0.018 |

**Additional ablation study results.** We report ablation study results on Spotify$_{k=30}$ (A=1 or A=10) in Table 9. The results obtained at A=10 are consistent with those observed at A=1. (1) The Residual

---

**Algorithm 3** OAS via Hungarian Algorithm (maximize sum of cosine similarities)

---

**Require:** Predicted set $\hat{\mathbf{b}}_y = \{\hat{i}_1, \ldots, \hat{i}_{\hat{n}}\}$ (duplicates removed); ground-truth set $\mathbf{b}_y = \{i_1, \ldots, i_n\}$; embeddings $E(\cdot)$
**Ensure:** Optimal matching $P \subseteq \{1, \ldots, \hat{n}\} \times \{1, \ldots, n\}$ and OAS
 1: **build similarity:** $S \in \mathbb{R}^{\hat{n} \times n}$ with $S[a, b] \leftarrow \cos\big(E(\hat{i}_a), E(i_b)\big)$
 2: $m \leftarrow \max(\hat{n}, n)$
 3: **build square cost:** $\tilde{C} \in \mathbb{R}^{m \times m}$          $\triangleright$ convert max-similarity to min-cost
 4: **for** $a = 1$ to $m$ **do**
 5:     **for** $b = 1$ to $m$ **do**
 6:         **if** $a \le \hat{n}$ **and** $b \le n$ **then**
 7:             $\tilde{C}[a, b] \leftarrow 1 - S[a, b]$          $\triangleright$ cost $\in [0, 2]$ since cos $\in [-1, 1]$
 8:         **else**
 9:             $\tilde{C}[a, b] \leftarrow 1$          $\triangleright$ dummy pairs have similarity 0
10:         **end if**
11:     **end for**
12: **end for**
13: **row reduction:** $\tilde{C}[a, \cdot] \leftarrow \tilde{C}[a, \cdot] - \min_b \tilde{C}[a, b]$ for all $a$
14: **column reduction:** $\tilde{C}[\cdot, b] \leftarrow \tilde{C}[\cdot, b] - \min_a \tilde{C}[a, b]$ for all $b$
15: **repeat**
16:     Cover all zeros in $\tilde{C}$ by the minimum number of horizontal/vertical lines
17:     **if** (#lines $< m$) **then**
18:         $\Delta \leftarrow \min\{\tilde{C}[a, b] : \tilde{C}[a, b]$ is uncovered$\}$
19:         Subtract $\Delta$ from every *uncovered* entry
20:         Add $\Delta$ to every *doubly-covered* entry
21:         (singly-covered entries unchanged)
22:     **end if**
23: **until** #lines $= m$
24: **extract assignment:** find $m$ independent zeros (no two share a row/column) to form an optimal assignment $\tilde{P} \subseteq \{1, \ldots, m\}^2$
25: **restrict to real items:** $P \leftarrow \{(a, b) \in \tilde{P} : a \le \hat{n}, \ b \le n\}$
26: $S_{\text{sum}} \leftarrow \sum_{(a,b) \in P} S[a, b]$
27: OAS $\leftarrow \dfrac{S_{\text{sum}}}{n}$          $\triangleright$ higher is better
28: **return** $P$, OAS

---

Vector Quantization (RVQ) component exhibits to be absolutely indispensable. This result confirms the substantial mechanism of RVQ to mitigate the dimensionality curse caused by the huge item catalog size (N). (2) The boi token could provide positional guidance and improve latent space quality largely. Removing the boi token results in a significant performance degradation. (3) Simple data augmentation (item swapping) proves to be a beneficial technique for enhancing order-agnostic modeling and improving model robustness by mitigating overfitting to specific bundle arrangements. (4) Although the Token Validity Filter yields only a marginal performance improvement, its inclusion remains necessary to guarantee the validity of the generated bundles during inference.

Table 9: Ablation study on Spotify$_{k=30}$ (A=1 vs. A=10). "Proposed" is our model DDBC; each of the other variants changes exactly one component that is removed from the proposed method.

| Variant | A=1 | | | A=10 | | |
|---|---|---|---|---|---|---|
| | F1 ↑ | Jacc ↑ | OAS ↑ | F1 ↑ | Jacc ↑ | OAS ↑ |
| Our proposed DDBC | 0.282 | 0.177 | 0.660 | 0.166 | 0.092 | 0.620 |
| *w/o RVQ* | 0.021 | 0.011 | 0.557 | 0.028 | 0.015 | 0.556 |
| *w/o token validity filter* | 0.276 | 0.173 | – | 0.163 | 0.090 | – |
| *w/o boi token* | 0.176 | 0.104 | 0.538 | 0.116 | 0.063 | 0.536 |
| *w/o data augmentation* | 0.254 | 0.158 | 0.599 | 0.152 | 0.084 | 0.598 |

Table 10: Effect of diffusion steps $T$ on Spotify$_{k=30}$.

| $T$ | F1 ↑ | Jacc ↑ | OAS ↑ |
|---|---|---|---|
| 1 | 0.2693 | 0.1656 | 0.6578 |
| 2 | 0.2713 | 0.1673 | 0.6571 |
| 4 | 0.2783 | 0.1724 | 0.6595 |
| 10 | 0.2798 | 0.1729 | 0.6604 |
| 16 | 0.2821 | 0.1753 | 0.6604 |
| 25 | 0.2821 | 0.1756 | 0.6600 |
| 32 | **0.2851** | 0.1771 | 0.6616 |
| 50 | 0.2850 | 0.1771 | **0.6618** |
| 75 | 0.2845 | **0.1773** | **0.6618** |
| 100 | 0.2832 | 0.1759 | 0.6595 |
| 200 | 0.2815 | 0.1745 | 0.6604 |
| 300 | 0.2816 | 0.1749 | 0.6601 |

Table 11: Impact of observed-item selection strategy on Spotify$_{k=30}$ ($A = 1$).

| Strategy | F1 ↑ | Jacc ↑ | OAS ↑ |
|---|---|---|---|
| Prefix | 0.2821 | 0.1756 | 0.6600 |
| Suffix | 0.2777 | 0.1717 | 0.6575 |
| Random | 0.2791 | 0.1727 | 0.6587 |

## C.1 EFFECT OF DIFFUSION STEPS

We further study how the number of reverse diffusion steps $T$ affects performance. All results are reported on Spotify$_{k=30}$ under the default evaluation protocol with $A = 1$. As shown in Table 10, increasing $T$ consistently improves performance from very small values, but the gain quickly saturates beyond a moderate range. Based on this trade-off between effectiveness and efficiency, we use $T = 25$ in the main experiments.

The results confirm an intuitive trend: too few denoising steps under-utilize the diffusion process, while excessively large $T$ yields only marginal improvements. This supports our choice of a moderate default horizon in the main paper.

## C.2 ROBUSTNESS TO THE CHOICE OF OBSERVED ITEMS

In the main experiments, we use the front part of a bundle as the observed subset $\mathbf{b}_x$, which ensures a fair comparison with sequential baselines trained under a left-to-right objective. To test whether DDBC is sensitive to this choice, we additionally evaluate three selection strategies for the observed items: **Prefix** (the first half of the bundle), **Suffix** (the second half), and **Random** (a random $50\%$ subset).

Table 11 shows that DDBC remains highly stable across all three settings, with only minor variations in F1, Jaccard, and OAS. This suggests that the model is not strongly tied to a particular observed-item ordering and is robust to different partial-bundle configurations. Minor fluctuations are still plausible, as some bundles may contain weak local sequential regularities.

## C.3 TRAINING-FREE VARIABLE-LENGTH GENERATION

Although the main paper focuses on a fixed-length evaluation protocol, DDBC can be adapted to variable-length generation without retraining. In this subsection, we report three simple inference-time strategies on Spotify$_{k=30}$ using the same trained model.

**Direct variable-length decoding.** We first directly change the number of item slots to be predicted from the fixed training length to a new target length bs. The results in Table 12 show that the model maintains competitive performance across a reasonably wide range of target lengths. In particular, when bs is close to the training length, the degradation is negligible.

Table 12: Direct variable-length generation on Spotify$_{k=30}$ without retraining.

| Target length bs | Recall ↑ | Precision ↑ | F1 ↑ | Jacc ↑ |
|---|---|---|---|---|
| 15 | 0.1550 | **0.3099** | 0.2066 | 0.1212 |
| 20 | 0.2002 | 0.3003 | 0.2403 | 0.1449 |
| 25 | 0.2004 | 0.3006 | 0.2405 | 0.1449 |
| 30 | 0.2821 | 0.2821 | 0.2821 | 0.1756 |
| 35 | 0.2415 | 0.2898 | 0.2635 | 0.1613 |
| 40 | **0.3878** | 0.2585 | **0.3102** | **0.1950** |
| 45 | 0.3868 | 0.2579 | 0.3094 | 0.1942 |

Table 13: Semi-AR variable-length generation on Spotify$_{k=30}$ without retraining.

| $R$ | bs | F1 ↑ | Jacc ↑ |
|---|---|---|---|
| 30 | 1 | 0.1244 | 0.0689 |
| 10 | 3 | 0.2133 | 0.1261 |
| 6 | 5 | 0.2656 | 0.1632 |
| 5 | 6 | 0.2772 | 0.1716 |
| 3 | 10 | **0.2837** | **0.1761** |
| 2 | 15 | 0.2825 | 0.1755 |
| 1 | 30 | 0.2821 | 0.1756 |

Table 14: Multi-sampling variable-length generation on Spotify$_{k=30}$ without retraining.

| $R$ | bs | F1 ↑ | Jacc ↑ |
|---|---|---|---|
| 30 | 1 | 0.1857 | 0.1101 |
| 15 | 2 | 0.1871 | 0.1111 |
| 10 | 3 | 0.2700 | 0.1687 |
| 6 | 5 | 0.2837 | 0.1755 |
| 5 | 6 | 0.2848 | 0.1768 |
| 3 | 10 | **0.2871** | **0.1782** |
| 2 | 25 | 0.2851 | 0.1770 |
| 1 | 30 | 0.2821 | 0.1756 |

**Semi-AR blockwise decoding.** Inspired by blockwise generation, we also consider a simple semi-autoregressive (Semi-AR) strategy. At each pass, the model predicts bs new items conditioned on the observed bundle $\mathbf{b}_x$ and the items generated so far; this process is repeated for $R$ passes. Table 13 shows that, as long as the per-pass block size is not too small, the performance remains close to the default fixed-length setting.

**Multi-sampling with set merging.** Finally, we test a non-Semi-AR alternative: draw multiple completions, merge the generated items into a candidate set, and evaluate the merged result against the target bundle. As shown in Table 14, this strategy leads to similar conclusions and is often effective when moderate over-generation is allowed.

**A note on block size.** We observe that using a moderate block size (roughly bs $\in [5, 10]$) can yield slightly better performance than predicting the entire bundle in a single shot. We conjecture that this may be because a bundle can contain multiple sub-themes, and generating items in several paced rounds may encourage covering different sub-themes across rounds rather than collapsing onto a single dominant mode. This hypothesis is preliminary, and we leave a more principled analysis to future work; it may also have implications for sequential recommendation, where multi-interest structure is commonly observed.

Overall, these preliminary results suggest that DDBC is not inherently tied to a single fixed output length. Even though the model is trained under a fixed-length protocol, it can still support variable-length generation through lightweight inference-time modifications, without any retraining. A more principled treatment of stopping criteria and length control remains an important direction for future work.

## D  FURTHER DISCUSSION

### D.1  FROM-SCRATCH BUNDLE GENERATION

Although our main evaluation follows the standard partial-to-full completion protocol, DDBC also defines an unconditional bundle prior and can generate bundles from scratch by initializing all item-code positions in $\Omega_{code}$ as [MASK] (i.e., $\mathbf{b}_x = \emptyset$) and running the same constraint-aware decoding in Algorithm 2. This capability is naturally supported by our absorbing-mask diffusion training, which learns to recover clean tokens from heavily corrupted states and thus exposes the denoiser to near-

unconditional inputs; in practice, all-mask samples tend to reflect global co-occurrence structure (frequent compatibilities are more likely to be generated jointly) and exhibit approximate order-insensitivity due to symmetric corruption. That said, without explicit conditioning, from-scratch sampling may drift toward popular or generic bundles, and variable-length generation still lacks principled stopping/length control; while token validity guarantees catalog validity, it does not guarantee user-specific relevance, motivating personalization and length-control extensions as future work.

## D.2 PERSONALIZED BUNDLE GENERATION

Our current instantiation focuses on unconditional completion with personalization primarily mediated by frozen encoders for user–item signals and item semantics. In real deployments, however, bundle generation is rarely signal-free: systems typically have intent cues (e.g., theme/genre/category) and user-specific preference signals, and the generated bundle should align with them. A lightweight, training-free way to incorporate such signals is (i) *seed–then–complete*: use an existing user-conditioned recommender (similarity, rules/filters, or profile-based recall) to retrieve a small set of seed items and then let DDBC expand it into a coherent full bundle; and (ii) *logit bias / external guidance*: during denoising, add a bias from an external user-conditioned score (e.g., user embedding similarity or category priors) to steer sampling while keeping the token-validity constraints. A more principled direction is to make the diffusion process explicitly conditional (e.g., injecting user/context features into the denoiser and training under the same masked objective), which could better trade off relevance, diversity, and controllability under personalization.

## D.3 CANDIDATE-POOL EVALUATION: RATIONALE AND LIMITATIONS

We evaluate under a controlled candidate pool rather than over the full item universe, and this choice is motivated by three pragmatic considerations. First, it ensures fair comparability: many strong baselines in bundle construction/recommendation are inherently candidate-conditioned (e.g., retrieval-then-rank or candidate-scoring pipelines), so removing the candidate pool would make comparisons ill-defined or require ad-hoc changes that alter computational budgets. Second, it partially addresses a real-world non-uniqueness issue: bundle construction often admits multiple valid solutions, and logged ground-truth bundles typically reflect only one feasible realization among many; near-neighbor substitutions (items that are semantically interchangeable) may yield equally coherent bundles yet be absent from the collected data, a recurring evaluation challenge in recommender systems beyond bundling. Third, it improves statistical reliability: in the fully open-world setting, exact-match hit rates can become extremely small for all methods, making results dominated by random variation and undermining meaningful comparisons; restricting to a shared candidate pool keeps success probabilities in a measurable range and reduces variance. We view this protocol as a reproducible compromise for benchmarking, while recognizing it does not fully capture open-world solution diversity; developing evaluation procedures that better credit multiple valid completions (e.g., graded relevance or set-level equivalence) remains important future work.

