# OpenReview forum: "Discrete Diffusion for Bundle Construction"
_ICLR.cc/2026/Conference — ICLR 2026 Poster_

### Official Review · Reviewer_CnC2 · 2025-10-29

**Soundness:** 2
**Presentation:** 2
**Contribution:** 3
**Rating:** 6
**Confidence:** 4

**Summary:**

Bundle construction suffers from a massive item pool and exponentially growing combinations.
DDBC addresses these challenges by utilizing a diffusion model to generate an entire bundle.
DDBC also quantizes item embeddings into discrete codes to reduce the search space.
Extensive experiments show that DDBC outperforms current bundle generation methods and achieves the state-of-the-art performance.

**Strengths:**

* Tokenizing items through codebook quantization is effective for processing a massive item pool.
* DDBC outperforms existing baselines by a huge margin.

**Weaknesses:**

* The size of bundle is fixed and not adjustable, unlike other baselines.
* Using the same item features for all baselines is doubtful. Each model may require its own embedding space depending on its architecture.

**Questions:**

* How does a STE work for a codebook quantization?
* How does the RVQ strategy illustrate an item in coarse-to-fine manner?
Sharing the same codebook for all positions without scaling seems to treat all tokens equally rather than hierarchically.
* Even if each item is quantized into a unique code sequence, it seems permutations of the sequence would represent semantically same features since dequantizing the residual quantization is aggregating all codes of the sequence.
How does this affect the diffusion process?
Is there a way to utilize this during the diffusion process?
* How does the dedup code working? Does it included in the codebook? Should the diffusion model recover the dedup code by denoising?

**Details Of Ethics Concerns:**

There is a name (subhamsahoo) that seems like an author in line 209 of ‘util.py’ in the code repository.
```
class BinarySampler(GumbelSampler):

  def sample(self, probs):
    # TODO(subhamsahoo): use the temperature parameter.
    pos_noise = self._sampling_noise().to(
      dtype=probs.dtype, device=probs.device)
    neg_noise = self._sampling_noise().to(
      dtype=probs.dtype, device=probs.device)
    del_noise_exp = (neg_noise - pos_noise).exp()
    hard_sample = (probs * (1 + del_noise_exp)
                   > 1).to(probs.dtype)
    soft_sample = probs / (probs + (1 - probs) * del_noise_exp)
    return soft_sample + (hard_sample - soft_sample).detach()
```

---

> ### Author Response · Authors · 2025-11-26
>
> We appreciate the insightful feedback and respond point-by-point below.
>
> **W1. The size of bundle is fixed and not adjustable.**
>
> Thank you for the concern. Since some baselines support retrieving an arbitrary number of items, while others are next-item generators that we roll out step by step. To keep the evaluation protocol uniform and the results comparable across all methods, we adopt a fixed $k$ for standardized evaluation.
>
> We also note that many prior methods[1, 2] are reported with fixed-cutoff metrics (e.g., Recall@10, NDCG@10), and next-item generators are typically rolled out to a target length; adopting a common $k$ keeps the protocol like-for-like.
>
> DDBC supports adjustable size without retraining. At inference, we set the number of masked slots (i.e., $\lvert \mathbf{b}_{y} \rvert$) to the desired target; the reverse diffusion then produces exactly that many items. For a more detailed discussion of variable bundle size and inference, please refer to our response to Reviewer B6yG, Q1.
>
> [1] Chang J, Gao C, He X, et al. Bundle recommendation with graph convolutional networks[C]//Proceedings of the 43rd international ACM SIGIR conference on Research and development in Information Retrieval. 2020: 1673-1676.
>
> [2] Ma Y, Liu X, Wei Y, et al. Leveraging multimodal features and item-level user feedback for bundle construction[C]//Proceedings of the 17th ACM International Conference on Web Search and Data Mining. 2024: 510-519.
>
> **W2. Using the same item features for all baselines is doubtful.**
>
> We appreciate the concern.
>
> This work focuses on comparing generative backbones for bundle construction. Therefore, we used the same encoder across methods to rule out confounding effects from the encoding stage. If each baseline used a different encoder (possibly with different objectives or pretraining data), performance differences would be harder to attribute to the generative mechanism rather than to feature quality. We did not intentionally weaken any baseline.
>
> Conceptually, different architectures correspond to different factorizations of the joint distribution, and the embedding spaces they are best suited to may indeed differ. In bundle scenarios, one often considers how to encode user–item, item–bundle, and bundle–user interactions, how to encode semantic signals from text/vision/audio, and how to fuse them.[1-3] It would be overly costly to exhaustively study encoder choices for each model.
>
> We view (1) learning latent representations of bundles/items and (2) designing the generative mechanism for bundle construction as two important but distinct questions. This paper focuses on the latter.
>
> [1] Chang J, Gao C, He X, et al. Bundle recommendation with graph convolutional networks[C]//Proceedings of the 43rd international ACM SIGIR conference on Research and development in Information Retrieval. 2020: 1673-1676
>
> [2] Ma Y, He Y, Zhang A, et al. CrossCBR: cross-view contrastive learning for bundle recommendation[C]//Proceedings of the 28th ACM SIGKDD conference on knowledge discovery and data mining. 2022: 1233-1241.
>
> [3] Ma Y, He Y, Wang X, et al. Multicbr: Multi-view contrastive learning for bundle recommendation[J]. ACM Transactions on Information Systems, 2024, 42(4): 1-23.
>
> **Q1. How does a STE work for a codebook quantization?**
>
> This STE stands for the Straight-Through Estimator[1]. In this work, train the RVQ codebooks using FAISS[2] , without modifying its underlying logic. Specifically, in the implementation of DDBC’s tokenizer:
>
> Given level-$l$ residual $r^{(l-1)} \in R^d$ and codebook $C^{(l)} = \{ e_c^{(l)} \}_{c=1}^{C_l}$, the forward assignment is
>
> $$
> z^{(l)}(i) = argmin_{c \in \{1,\ldots,C_l\}} \, \| r^{(l-1)} - e_c^{(l)} \|_2^2,
> $$
>
> $$
> r^{(l)} = r^{(l-1)} - e_{z^{(l)}(i)}^{(l)}.
> $$
>
> We implement it with a straight-through copy:
>
> $$
> \hat r^{(l-1)} = r^{(l-1)} + sg\big[ e_{z^{(l)}(i)}^{(l)} - r^{(l-1)} \big],
> $$
>
> so in the backward pass
>
> $$
> \frac{\partial \hat r^{(l-1)}}{\partial r^{(l-1)}} \approx I, \quad
> \frac{\partial \hat r^{(l-1)}}{\partial e_{z^{(l)}(i)}^{(l)}} \approx 0.
> $$
>
> Codebooks are trained with the RVQ loss (applied for $l=1,\ldots,L-1$):
>
> $$L\_{RVQ} = \| E(i) - \hat E(i) \|_2^2 + \beta \sum\_{l=1}^{L-1} \Big( \| sg[r^{(l-1)}] - e^{(l)}\_{z^{(l)}(i)} \|_2^2 + \| r^{(l-1)} -sg[e^{(l)}\_{z^{(l)}(i)}] \|_2^2 \Big),$$
>
> where
>
> $$
> \hat E(i) = \sum\_{l=1}^{L-1} e^{(l)}\_{z^{(l)}(i)}.
> $$
>
> [1] Bengio Y, Léonard N, Courville A. Estimating or propagating gradients through stochastic neurons for conditional computation[J]. arXiv preprint arXiv:1308.3432, 2013.
>
> [2] Douze M, Guzhva A, Deng C, et al. The faiss library[J]. IEEE Transactions on Big Data, 2025.

---

> ### Author Response · Authors · 2025-11-26
>
> **Q2. How does the RVQ strategy illustrate an item in coarse-to-fine manner?**
>
> The "coarse-to-fine" structure lies in the design of RVQ: an item’s semantic embedding produced by the CLHE encoder is decomposed by RVQ into a sequence of tokens, and these tokens encode semantic information in a coarse-to-fine manner [1, 2]. When two items are semantically similar, it is very likely that their first few tokens coincide.
>
> In other words, “coarse-to-fine” is a tokenization-level notion rather than a temporal/decoding/positional notion. Even without an explicit level-wise scaling factor in reconstruction, the hierarchical behavior emerges from the residual quantization: early levels account for the bulk of the norm; later levels refine residuals.
>
> [1] Rajput S, Mehta N, Singh A, et al. Recommender systems with generative retrieval[J]. Advances in Neural Information Processing Systems, 2023, 36: 10299-10315.
>
> [2] Kim J, Moon T, Lee K, et al. Efficient Generative Modeling with Residual Vector Quantization-Based Tokens[J]. arXiv preprint arXiv:2412.10208, 2024.
>
>
> **Q3. Is there a way to utilize the permutation-ability of item's token sequence?**
>
> We agree that explicitly exploiting the approximate symmetry over level permutations is an interesting direction. However, such designs would require non-trivial changes to the decoder architecture and training objective, and are beyond the scope of this work.
>
> Intuitively, as long as each item slot is composed of four tokens drawn from four different levels, permutations of the level indices could indeed lead to identical reconstructed embeddings.
>
> In our implementation, however, RVQ levels are designed *not* interchangeable: each position in the $L$-tuple corresponds to a specific RVQ level with its own codebook and residual semantics. The reason is that:
>
> Even if we deliberately treated level permutations as equivalent, allowing arbitrary permutations during training would create a many-to-one mapping between token sequences and items. With 4 levels, each item would effectively have $4!$ valid permutations of its code tuple. For a token-level generative model trained with cross-entropy (autoregressive or diffusion-based), this leads to ambiguous supervision: multiple distinct sequences would all be “correct” targets for the same item. This ambiguity tends to slow convergence and degrade generation quality, and is at odds with the standard language-model training paradigm where each context has a single canonical next-token label.
>
>
> **Q4a. How does the dedup code working?**
>
> As defined in Section 3.2, we apply an $L$-level RVQ to obtain
>
> $$
> z(i) = \big( z^{(1)}(i), \ldots, z^{(L)}(i) \big)
> $$
>
> for each item $i$, where the last level $z^{(L)}(i)$ is a *dedup code* that carries no semantics and acts purely as an auto-increment field to ensure a one-to-one mapping from a code tuple back to a unique item ID.
>
> Formally, $z^{(l)}(i) \in \{1,\ldots,C\_l\}$ indexes a codeword in codebook $C^{(l)}$, but the reconstruction
>
> $$
> E\_hat(i) = \sum\_{l=1}^{L-1} e^{(l)}\_{z^{(l)}(i)}
> $$
>
> and the RVQ loss $L\_{RVQ}$ only involve levels $1,\ldots,L-1$. In practice, we first train the semantic levels $1,\ldots,L-1$, then assign distinct $z^{(L)}(i)$ within each collision group to break ties.
>
>
> **Q4b. Does it included in the codebook?**
>
> Yes. For the diffusion model, the dedup code is treated like any other discrete token: it is included in the vocabulary and is recovered by denoising together with the semantic levels. This keeps the representation simple and consistent: every item is always represented by an $L$-tuple, and the diffusion model learns to reconstruct the full tuple.
>
>
> **Q4c. Should the diffusion model recover the dedup code by denoising?**
>
> We believe that our current choice of letting the diffusion model recover the dedup code by denoising is reasonable. As shown in Table 4, we also experimented with removing the dedup code and observed a drop in performance.
>
> Collisions are not unusual, so the dedup code guarantees a unique mapping from denoised token sequences back to catalog items. Without the dedup code, it would be equivalent to ignoring many items, especially long-tail items, since they would be merged.
>
> Intuitively, most of the information comes from the first $L-1$ levels. We believe that, when designing the decoding strategy, one could first predict $z^{(1)},\ldots,z^{(L-1)}$. This would not change the semantics or evaluation, and we view it as a potential practical decoding strategy of the work.
>
>
> **To Ethics Review**
>
> Thank you for pointing this out. Our implementation builds on the open-source MDLM repository. The specific comment in question originates from the original authors of MDLM (https://github.com/kuleshov-group/mdlm/blob/master/utils.py line 209). We will update the README to clearly credit MDLM as the backbone and specify the license.

---

### Official Review · Reviewer_ZPfg · 2025-11-01

**Soundness:** 2
**Presentation:** 3
**Contribution:** 1
**Rating:** 4
**Confidence:** 3

**Summary:**

In this paper, the authors propose DDBC (Discrete Diffusion for Bundle Construction), a novel framework that formulates bundle construction as a masked discrete diffusion process. Rather than following traditional step-by-step methods, DDBC generates bundles in an order-independent way by progressively filling in masked item tokens. The framework combines Residual Vector Quantization (RVQ), which represents items with discrete semantic codes to address the challenge of large item catalogs, with a Discrete Diffusion Model (DDM) designed to capture complex relationships within bundles without requiring a fixed item order. Comprehensive experiments on Spotify playlist and fashion outfit datasets show that DDBC achieves substantial performance improvements over both sequential and set-based baselines. Ablation studies further highlight the complementary strengths of RVQ and the discrete diffusion backbone.

**Strengths:**

1. The paper presents a well-motivated argument against the sequential construction paradigm, emphasizing that bundles are inherently unordered sets rather than sequences.
2. The integration of discrete diffusion and residual vector quantization is technically coherent and aligns with recent advances in generative recommendation.
3. The paper provides extensive experiments with clear ablations and sensitivity analyses, showing that DDBC scales better with longer bundles and larger catalogs.

**Weaknesses:**

1. While the paper demonstrates robustness across different input-predict ratios, the framework fundamentally requires at least some partial bundle items at inference. The paper does not explore conditional generation from fully-masked states using only user embeddings or contextual features, which limits applicability for cold-start scenarios where no seed items are available.
2. The model enforces a fixed number of items per bundle, whereas real-world bundles are inherently variable in size. Capping bundle lengths during training and testing introduces a methodological limitation that does not fully capture the characteristics of the bundle item collection as in original dataset.
3. While the integration of RVQ with discrete diffusion is technically sound and well-executed, the core novelty lies primarily in combining existing techniques (RVQ from generative retrieval and masked diffusion from language models) rather than introducing fundamentally new algorithms. The ablation study demonstrates that both components are essential and complementary, but the contribution is more engineering-focused than conceptually novel. The paper would benefit from deeper theoretical analysis of why this combination works particularly well for bundle construction.

Minor typos:
-	L122: Citation formatting error
-	L196: Use $z_{j,l}$ instead of $z_{jl}$
-	L244: The reverse process should be conditioned on current noisy tokens $Z^{(t)}$
-	L267: $\hat{E}_j = \sum_{l=1}^{L-1} e^{(l)}_{z^{(l)}(i)}$
-	L235, L287: The notation $\alpha(t) != \alpha_t$; $\alpha(t)$ should be $1-t/T$ to yield deterministic at $t=T$

**Questions:**

1. At $t=T$, all tokens are fully masked, yet the model is trained to predict original tokens under Equation 7. While this teaches the model unconditional bundle priors, could the authors clarify:
    - a) What proportion of training loss comes from high-noise timesteps ($t \geq 0.8T$)
    - b) Whether this supervision is necessary given that inference always starts from partially-observed bundles
    - c) Whether alternative training strategies (e.g., restricting $t < T$) were explored?
2. During inference, how are the “known” items in $b_x$ selected? Since bundle semantics could vary depending on which subset is revealed, does the model show robustness to different partial-item configurations?
3. When bundles exceed the capped size $k$, how are items selected for truncation? Does this selection (e.g., first-k, random-k, most-popular-k) introduce bias toward specific bundle patterns? Have the authors analyzed whether truncation affects semantic coherence or diversity of the resulting training data?

---

> ### Author Response · Authors · 2025-11-26
>
> We thank the reviewer for the time and effort devoted to assessing our work, and for the insightful remarks and helpful guidance.
>
> **W1. While the paper demonstrates robustness across different input-predict ratios, the framework fundamentally requires at least some partial bundle items at inference. The paper does not explore conditional generation from fully-masked states using only user embeddings or contextual features, which limits applicability for cold-start scenarios where no seed items are available.**
>
> We appreciate the reviewer’s concern. Our problem formulation follows prior literature~[1], which defines bundle construction given partially observed bundle items.
>
> We would like to highlight that the DDBC backbone is trained in an unconditional manner over complete bundles: the forward process corrupts full bundles to the all-mask state, and the reverse process learns to reconstruct them. Thus, at inference time the model is not restricted to partial bundles; it can also start from an all-mask state and sample an entire bundle from the learned prior without any seed items.
>
> We view *Personalized Bundle Construction* (conditioning on user embeddings or other contextual features) as a valuable but orthogonal direction; it is not the primary task that DDBC aims to solve. However, as discussed in our responses to Reviewer UnGS (Weakness 3), we outline training-free adaptation methods for personalization.
>
> [1] Ma Y, Liu X, Wei Y, et al. Leveraging multimodal features and item-level user feedback for bundle construction. In *WSDM*, 2024: 510–519.
>
> **W2. The model enforces a fixed number of items per bundle, whereas real-world bundles are inherently variable in size. Capping bundle lengths during training and testing introduces a methodological limitation that does not fully capture the characteristics of the bundle item collection as in original dataset.**
>
> We agree that real-world bundles are variable in size. In this work we fix the target length primarily to control confounders and ensure fair comparison across baselines. This choice does not alter the modeling objective, which targets the joint distribution over bundle items for a given cardinality.
>
> In this work, we do not predict an explicit *end-of-bundle* signal. In practice (e.g., music/movies/games), bundle growth rarely exhibits a clear stopping event and may evolve over time. However, DDBC supports variable-length generation with training-free inference protocols (Direct, Semi-AR, and Multi-sampling), for which we report preliminary results in Reviewer B6yG, Q1. In deployment, the final length can be decided by downstream policies (UI pagination, or user feedback), while the model focuses on generating high-quality items.
>
> A more detailed explanation and analysis of the limitation and rationale of capping bundle lengths can be found in our response to your Q3.
>
> We acknowledge that modeling variable-length bundles is valuable. For example, a semi-causal design could learn when to emit a `<pad>` or `<stop>` signal; diffusion-forcing–style mechanisms may also be relevant.[1] We view this as an interesting direction for future work.
>
> [1] Chen B, Martí Monsó D, Du Y, et al. Diffusion forcing: Next-token prediction meets full-sequence diffusion[J]. Advances in Neural Information Processing Systems, 2024, 37: 24081-24125.

---

> ### Author Response · Authors · 2025-11-26
>
> **W3. The paper would benefit from deeper theoretical analysis of why this combination works particularly well for bundle construction.**
>
> We appreciate the reviewer’s point. We will restate our motivation and explain why we use discrete diffusion to model bundle construction rather than an autoregressive approach.
>
> **Motivation**
>
> Our motivation is to address set-level bundle construction under large catalogs, where (i) autoregressive orderings impose exposure bias for an exchangeable target, and (ii) continuous spaces make validity (i.e., decoding to a legal catalog item) difficult at scale. Our technical contribution is to combine residual vector quantization with masked discrete diffusion, so that (a) items are represented as short code tuples with a coarse-to-fine structure, (b) the diffusion process is order-agnostic at the set level, and (c) each generated $L$-tuple deterministically decodes to a valid item.
>
> **AR vs Diff**
>
> From a theoretical perspective, our modeling targets the *joint distribution of a bundle* $p\_{\text{data}}(b)$ (with fixed cardinality $\|b\|$).
>
> **AR factorization and exposure bias.**
>
> Pick a permutation $\pi$ and define
>
> $$
> p_\phi(b)
> = \prod_{t=1}^{|b|} p_\phi\big(i_{\pi_t}\,\big|\, i_{\pi_{<t}}\big).
> \tag{1}
> $$
>
> Teacher forcing minimizes the empirical risk
>
> $$
> \mathcal{L}_{\mathrm{AR}}(\phi)
> = \mathbb{E}\_{\pi} \mathbb{E}\_{b \sim p\_{\mathrm{data}}}
> \left[
> -\sum\_{t=1}^{|b|}\log p\_\phi(i\_{\pi\_t}\mid i\_{\pi\_{< t}})
> \right]
> \tag{2}
> $$
>
>
> At test time, prefixes are drawn from the model rather than $p_{\text{data}}$, creating a distributional mismatch that can be summarized as an average KL gap over prefix distributions,
>
> $$
> \Delta_{\mathrm{exp}}
> = \frac{1}{|b|}\sum_{t=1}^{|b|} \mathrm{KL}\left(
> P_{\text{prefix}}^{\text{train}}(\cdot)\,\big\|\,P_{\text{prefix}}^{\text{test}}(\cdot)
> \right),
> \tag{3}
> $$
>
> which typically grows with $t$ as later items condition on more potentially erroneous model tokens. Moreover, any single $\pi$ imposes an order; randomizing $\pi$ merely averages (1) over orders rather than directly optimizing an order-insensitive set likelihood.
>
> **Discrete diffusion.**
>
> Serialize $b$ into a length-$|b|$ vector $x_0=(x_{0,1},\dots,x_{0,|b|})$ of item identifiers (arbitrary order). Let $q_t(x_t\mid x_0)$ independently mask each position with rate $\beta_t$ (absorbing mask). For each coordinate $j\in\{1,\dots,|b|\}$ with survival $\alpha_t=\prod_{s=1}^{t}(1-\beta_s)$,
>
> $$
> q\big(x_{t,j}=v \,\big|\, x_{0,j}=u\big)
> = \alpha_t\,\mathbf{1}[v=u] + (1-\alpha_t)\,\mathbf{1}[v=\text{[MASK]}].
> \tag{4}
> $$
>
> The denoising objective maximizes the likelihood of original tokens under random corruption:
>
> $$
> \mathcal{L}\_{\mathrm{Denoise}}(\theta)
> = \mathbb{E}\_{t,\,x_0\sim p\_{\mathrm{data}},\,x_t\sim q_t}
> \left[
> -\sum\_{j\in M_t}\log p_\theta(x\_{0,j}\mid x_t, t)
> \right]
> \tag{5}
> $$
>
> where $M_t$ is the random set of masked positions at step $t$. Because $M_t$ is sampled *uniformly over positions*, the gradient treats slots symmetrically:
>
> $$
> \nabla_\theta \mathcal{L}\_{\mathrm{Denoise}}
> = \mathbb{E}\_{M_t}
> \left[
> \sum\_{j\in M_t}
> \nabla_\theta\big(-\log p_\theta(x_{0,j}\mid x_t,t)\big)
> \right]
> \tag{6}
> $$
>
>
> yielding *order-insensitive training in expectation* under a position-symmetric corruption $q_t$. In practice, we use positional encodings to distinguish slots; this breaks exact permutation invariance, so the property above should be understood as an *approximate* order-insensitivity induced by symmetric masking and aggregated supervision.
>
> In summary, AR optimizes an ordered factorization; masked discrete diffusion optimizes an order-insensitivity denoising objective, naturally matching the task object.
>
> **RVQ**
>
> Compared to one-hot item IDs, RVQ shares code tokens across many items. Let $\mathcal{I}\big(z^{(\ell)}\big)=\{\, i : z^{(\ell)}(i)=z^{(\ell)} \,\}$. The expected token-level log-likelihood aggregates supervision:
>
> $$
> \mathbb{E}\_{i\sim p_{\mathrm{data}}}
> \left[
>   \log p_\theta(z^{(\ell)}(i)\mid x_t, t)
> \right]
> = \sum\_{k} p\_{\mathrm{data}}(\mathcal{I}(k))
> \log p\_\theta(k\mid\cdot)
> \tag{7}
> $$
>
> so rare items benefit from *shared code statistics*, while later RVQ levels focus on small residuals—an inductive bias for *compatibility refinement* inside bundles.
>
> Overall, we acknowledge that this is an application-oriented paper. It does not claim innovations on foundational aspects of diffusion models (e.g., model paradigms or training thermodynamics); rather, it examines how to choose a better factorization under the specific setting and datasets of bundle construction. We also welcome suggestions on theoretical angles the reviewers would find most valuable – please give us some hints.
>
>
> **Typos.**
>
> We thank the reviewer for carefully pointing out these typos. We will correct all of them in the revised version.

---

> ### Author Response · Authors · 2025-11-26
>
> **Q1a What proportion of training loss comes from high-noise timesteps ( $t\leq 0.8T$)**
>
> We sample $t$ uniformly from $\{1, \dots, T \}$ with an increasing mask schedule; thus steps with $t/T\ge 0.8$ constitute exactly $20\%$ of updates. The objective averages cross-entropy over both $t$ and tokens. We did not explicitly log the decomposition of the specific value of the total training loss by timestep.
>
> **Q1b Whether this supervision is necessary given that inference always starts from partially-observed bundles**
>
> Yes. Even though, at inference time, the tokens of the observed items are fully given, the tokens of the unobserved items are initialized from an all-mask state. High-noise (near–$T$) training steps teach the model to denoise from such fully masked positions, which directly matches the prediction setting for the missing items.
>
> In addition, including $t = T$ avoids a train–test mismatch for deployment scenarios where one may wish to initialize the entire bundle from an all-mask state.
>
> **Q1c Whether alternative training strategies (e.g., restricting $t \leq T$) were explored?**
>
> We did not experiment with strategies that explicitly restrict $t < T$ in the current version.
>
> We agree this is an interesting ablation: it could potentially reduce reliance on fully masked states but may also weaken the learned unconditional prior and robustness when denoising from heavily corrupted inputs. We plan to explore such variants and report the results in an updated respond.
>
> **Q2. During inference, how are the “known” items in bx selected?**
>
> We appreciate this insightful question. In our evaluation, we provide the *front* items of the bundle as the “known” items, as illustrated in Fig.~1. This choice ensures a fair comparison with sequential baselines: they are trained to generate left-to-right, and reversing or randomizing the known subset would place them at a disadvantage due to their training objective.
>
> To assess robustness, we tested three selection strategies: using the Prefix (first half) as input, the Suffix (second half) as input, and a Random (random sample 50\% of items) as input. The results show that DDBC is robust across these configurations, with only small variations.
>
> Table: Impact of “known” items selection strategy on performance
>
> | Strategy | F1     | jaccard | OAS   |
> |----------|--------|---------|-------|
> | Prefix   | 0.2812 | 0.1743  | 0.34  |
> | Suffix   | 0.2777 | 0.1717  | 0.3425|
> | Random   | 0.2791 | 0.1727  | 0.3413|
>
> The deltas are small (e.g., $\Delta\mathrm{F1}\le 0.004$, $\Delta\mathrm{Jaccard}\le 0.003$), indicating that DDBC is insensitive to which half is observed. Minor differences are plausible because some bundles exhibit weak sequential dependencies (as mentioned in the paper (line~50)), but their impact is limited.
>
> We recognize that this robustness test is most informative when contrasted with sequential baselines whose training objectives are prefix-dependent.
> We will update our respond, once we analyze sequential baselines under the same swapping protocol and provide further discussion.
>
> **Q3a. When bundles exceed the capped size k, how are items selected for truncation?**
>
> For $|b| \geq k$, we select a *random contiguous block of length $k$ without wrap-around*: we sample a start index
>
> $$
> s \sim \mathrm{Uniform}\{1,\dots,|b|-k+1\},
> $$
>
> and retain
>
> $$
> S=\{\,i_s,\,i_{s+1},\,\dots,\,i_{s+k-1}\,\}.
> $$
>
> **Q3b. Does this selection (e.g., first-k, random-k, most-popular-k) introduce bias toward specific bundle patterns?**
>
> Yes—there can be some bias if we assume very long-range dependencies among bundle items.
>
> Among the strategies you listed, a most-popular-$k$ rule would clearly introduce popularity bias. Our policy can be viewed as a hybrid between first-$k$ and random-$k$: like first-$k$, we keep a contiguous segment; like random-$k$, the start position is randomized. This avoids prefix bias while inducing a locality bias, which we consider reasonable for playlists/outfits where local order reflects editorial structure. In addition, we expect the practical impact of different truncation policies to be small: during training, item tokens are randomly corrupted in the forward process, naturally creating rich, non-contiguous contexts in which each item has a comparable chance to be predicted—effectively covering the random-$k$ case as well.
>
> **Q3c. Have the authors analyzed whether truncation affects semantic coherence or diversity of the resulting training data?**
>
> No we haven't, and we appreciate this perspective.
> Intuitively, keeping contiguous blocks tends to benefit coherence (local structure is preserved) but may reduce global diversity on very long bundles.
>
> However, we cannot guarantee that we will be able to design and evaluate a no-truncation setting for comparison before the rebuttal deadline. A more comprehensive study of truncation–diversity trade-offs is valuable and will be left to future work.

---

### Official Review · Reviewer_B6yG · 2025-11-05

**Soundness:** 3
**Presentation:** 3
**Contribution:** 2
**Rating:** 6
**Confidence:** 3

**Summary:**

The paper proposes DDBC, which is a discrete diffusion for bundle construction. The core motivation is that pervious bundle construction methods are based on sequential generation, where bundle length grows results in intra-bundle relational explosion and large catalog size makes item search space massive. DDBC treats bundle construction as masked discrete denoising over a compact discrete code space. DDBC quantizes item embeddings via multi-level RVQ, running masked discrete diffusion over the codes, and decoding the codes back to item IDs. The experiments show significant gains (about 100%) on some of the baselines.

**Strengths:**

The bundles are regarded as sets not sequences, which is . And the combination of the masked discrete diffusion and RVQ for bundle generation is novel in this domain.

The improvements are significant compared to the baselines. DDBC achieves >100% relative gains in Jaccard and F1 on Spotify with k = 60/90. DDBC scales well to have bigger relative improvements on the datasets with larger k.

DDBC is tiny and cost less inference time compared to many baselines like BundleMLLM.

**Weaknesses:**

The size of bundle seems to be fixed-length. If so, the application of DDBC could be limited. DDBC does not model set invariance. Although the bundles are not designed as serialized sequences and order is randomized, permutation invariance is not considered.

The evaluation includes only playlist and POG. I wonder if DDBC still performs well when bundles are from completely different domain, e.g., shopping sets from different categories.

The way of handling personalization is not explored. The generation is unconditional w.r.t user preferences, e.g., the RVQ codebooks of DDBC are global, potentially flattening user preference patterns. The current framework may hardly be deployed to scenarios where personalization is required.

**Questions:**

Could the diffusion be modified to support variable-length bundle reconstruction?

Could DDBC extended to personalization scenarios? e.g., change architecture to conditional diffusion and inject the user preference using the conditions.

---

> ### Author Response · Authors · 2025-11-26
>
> We appreciate the reviewer’s thoughtful comments.
>
> **W1a. The size of bundle seems to be fixed-length.**
>
> In our current setting, we consider fixed-size bundles to ensure fair evaluation; however, with simple inference-time modifications, we can also adapt to variable-length bundles in a training-free manner, with only a minor drop in accuracy. We conducted preliminary experiments on this, and the details are provided in our response to your Question 1.
>
> **W1b. Permutation invariance is not considered.**
>
> Thank you for raising this concern.
> We do not explicitly enforce strict permutation invariance, which we state in the paper.
> Instead, we approximate set-like behavior through our absorbing-mask diffusion design: during training, each time item tokens are randomly corrupted in the forward process, creating rich, non-contiguous contexts.[1, 2] This encourages the model to rely on the multi-set of items rather than absolute positions, since any slot can be masked or revealed over the trajectory.
>
> Exploring fully permutation-invariant diffusion backbones (e.g., set- or graph-based variants) is complementary to our current design and left for future work.
>
> [1] Austin J, Johnson D D, Ho J, et al. Structured denoising diffusion models in discrete state-spaces[J]. Advances in neural information processing systems, 2021, 34: 17981-17993.
>
> [2] Sahoo S, Arriola M, Schiff Y, et al. Simple and effective masked diffusion language models[J]. Advances in Neural Information Processing Systems, 2024, 37: 130136-130184.
>
> **W2. Should evaluate on different domain datasets.**
>
> We appreciate the suggestion to test additional domains.
> We do intend to include more datasets; however, datasets in other domains differ substantially in scale from Playlist and Fashion.
>
> Table: Common bundle datasets across domains. Counts are approximate and may vary by preprocessing.
>
> | Dataset  | Domain    | Item catalog size | # Bundles |
> |---------|-----------|-------------------|-----------|
> | Spotify | Playlists | $\sim$250K        | $\sim$300K |
> | POG     | Fashion   | $\sim$31K         | $\sim$29K |
> | NetEase | Playlists | $\sim$124K        | $\sim$23K |
> | Polyvore| Fashion   | $\sim$126K        | $\sim$21K |
> | Youshu  | Booklists | $\sim$32K         | $\sim$5K  |
> | Steam   | Games     | $\sim$3K          | $\sim$0.6K |
>
> Adding a new dataset entails several steps: extracting semantic identifiers, extracting collaborative-filtering signals, merging them into a unified item embedding, training the RVQ codebooks, and then training the model. We would appreciate it if the reviewers could recommend one or more suitable datasets and an encoder model. We will do our best to report results by the rebuttal deadline.
>
> **W3. Personalization (scope and practical extensions).**
>
> We agree that personalization is crucial. Our current instantiation focuses on unconditional completion; the Discussion explicitly proposes to add conditional diffusion (e.g., user/context features or instruction) as future work. However, our model is highly extensible: personalization can be added at inference time without retraining.
>
> Please refer to our response to Reviewer UnGS, Weakness 5, where we provided two extensions: (i) use a recommender to recall several user-preferred items as seeds and then expand the bundle; (ii) inject a bias during the denoising stage. We believe Personalized Bundle Construction is a distinct task with a different problem formulation.
>
> We believe Personalized Bundle Construction is a distinct task with a different problem formulation. It is valuable for future research and requires consideration of multiple practical details, which are beyond the scope of this paper.
>
> Also, we'd like to note that the RVQ codebooks encode item semantics, whereas personalization enters via conditioning/bias. Thus, global codebooks should not flatten user patterns once inference is guided.

---

> ### Author Response · Authors · 2025-11-26
>
> **Q1. Could the diffusion be modified to support variable-length bundle reconstruction?**
>
> We appreciate the reviewer for highlighting the importance of arbitrary-length generation in real applications. In this paper, our evaluation setting does not include this case. DDBC supports variable length with training-free modifications. To address your question, we conducted preliminary experiments and tried three arbitrary-length:
>
> First, we directly changed the number of tokens to be predicted. We observed that, although the model is trained in fix-length, within a wide range, the model maintains a high F1.
>
> Table: Direct arbitrary-length generation performance.
>
> | n' | Recall | Precision | F1     | Jaccard |
> |----|--------|-----------|--------|---------|
> | 15 | 0.1550 | 0.3099    | 0.2066 | 0.1212  |
> | 20 | 0.2002 | 0.3003    | 0.2403 | 0.1449  |
> | 25 | 0.2004 | 0.3006    | 0.2405 | 0.1449  |
> | 30 | 0.2821 | 0.2821    | 0.2821 | 0.1756  |
> | 35 | 0.2415 | 0.2898    | 0.2635 | 0.1613  |
> | 40 | 0.3878 | 0.2585    | 0.3102 | 0.1950  |
> | 45 | 0.3868 | 0.2579    | 0.3094 | 0.1942  |
>
> We further try a semi-AR decoding scheme similar to block diffusion[1]. At each pass, we generate $n'$ items, and repeat for $k'=\lvert b_y\rvert / n'$. At each pass, the model conditions on the context $b_x$ and the items generated so far. Finally, we concatenate all generated items and compare the result with $b_y$.
>
> We find that as long as the per-step block size is not too small (which would undercut the model’s capability), using $>5$ items per step yields consistent performance.
>
> Table: Semi-AR arbitrary-length generation performance.
>
> | k' | n' | F1     | Jaccard |
> |----|----|--------|---------|
> | 30 | 1  | 0.1244 | 0.0689  |
> | 10 | 3  | 0.2133 | 0.1261  |
> | 6  | 5  | 0.2656 | 0.1632  |
> | 5  | 6  | 0.2772 | 0.1716  |
> | 3  | 10 | 0.2837 | 0.1761  |
> | 2  | 15 | 0.2825 | 0.1755  |
> | 1  | 30 | 0.2821 | 0.1756  |
>
> Similarly, we also try a non–semi-AR variant: we input $b_x$ and draw multiple samples to obtain $k' \times n'$ candidates, merge them, and compare the merged set with $b_y$. We observe similar conclusions.
>
> Table: Multi-Sampling arbitrary-length generation performance.
>
> | k' | n' | F1     | Jaccard |
> |----|----|--------|---------|
> | 30 | 1  | 0.1857 | 0.1101  |
> | 15 | 2  | 0.1871 | 0.1111  |
> | 10 | 3  | 0.2700 | 0.1687  |
> | 6  | 5  | 0.2837 | 0.1755  |
> | 5  | 6  | 0.2848 | 0.1768  |
> | 3  | 10 | 0.2871 | 0.1782  |
> | 2  | 25 | 0.2851 | 0.1770  |
> | 1  | 30 | 0.2821 | 0.1756  |
>
> We will add a short protocol description and these numbers to the appendix, and clarify that variable-length generation is supported without retraining.
>
> [1] Arriola M, Gokaslan A, Chiu J T, et al. Block diffusion: Interpolating between autoregressive and diffusion language models[J]. arXiv preprint arXiv:2503.09573, 2025.
>
> **Q2. Could DDBC extended to personalization scenarios?**
>
> Yes. As discussed in our responses to Weakness 3 raised by Reviwer UnGS and Weakness 5 by you, we outline training-free adaptation methods (seed–then–complete and logit bias) for personalization.
>
> For *training-based* conditioning, we agree with your suggestion: DDBC can be made conditional by augmenting the denoiser with user/context features under the same masked discrete objective, and applying classifier-free guidance at sampling to trade off condition alignment and diversity. We recommend following the discrete-guidance implementation of [1].
>
> [1] Schiff Y, Sahoo S S, Phung H, et al. Simple guidance mechanisms for discrete diffusion models[J]. arXiv preprint arXiv:2412.10193, 2024.

---

### Official Review · Reviewer_UnGS · 2025-11-09

**Soundness:** 3
**Presentation:** 3
**Contribution:** 3
**Rating:** 8
**Confidence:** 4

**Summary:**

This paper proposes a discrete diffusion based bundle construction method. To address the complexity challenge of bundle construction caused by the bundle size and large item corpus, it (1) uses RVQ to get the code representation of each item by training a shared codebook; and (2) trains a discrete diffusion model to gradually mask items in a given bundle in the forward process and then gradually reconstruct the masked items in the reverse process with the item token learned through RVQ module. Experiments on two real-world dataset partially prove the effectiveness of the proposed method (as from the result, the proposed method only works well on one of the adopted dataset with larger bundle size). Overall, this is a quite solid study on bundle construction.

**Strengths:**

S1. The paper is generally well written and organized, which makes it easy to follow.

S2. The investigated research question is quite meaningful and practical in the real-world application. More importantly, the authors identified the limitations of the existing sequential-based bundle construction, as the order of the items within a bundle may not matter a lot when constructing the bundle (it could be a set).

S3. The effectiveness of the model has been verified on the Spotify dataset with the large bundle size, while it does not work on the POG dataset with smaller bundle size.

S4. Extensive ablation studies and important hyper-parameter analysis further show the efficacy of the important modules of the model

S5. The source code is provided to ensure a better reproducibility.

**Weaknesses:**

W1. The title does not really reflect the task that has been investigated in the paper. The paper actually investigates the bundle completion task instead of exact bundle construction task.

W2. It would better if the authors can highlight their technical contribution.

W3. The investigate the research question, though being practical and meaningful in real-world application, seems to lack generalizability. It relies on partial items in a bundle, so cannot create a bundle from scratch. However, in the real-world applications, there are many scenarios, where creating bundle from scratch is required.

W4. Some related works are missing, for instance, Adaptive In-Context Learning with Large Language Models for Bundle Generation (SIGIR 2024). The authors are highly suggested to cover a comprehensive literature, especially LLM based method for bundle construction/completion.

W5. The fixed-length setting may limit the flexibility of the proposed model. In the real-scenario, we may not be able to show too many items for the users in one page, so how to determine the subset of the items that should be displayed to the user?

**Questions:**

Q1. How to distinguish the tokens of unpopular items using RVQ? Existing study shows that RVQ in the Euclidean space may not be able to well distinguish the unpopular items, which has a large amount in the real-world scenarios.

Q2. Does the step T affect the performance? If so, how?

---

> ### Author Response · Authors · 2025-11-26
>
> We thank the reviewer for the careful assessment and constructive suggestions.
>
> **W1. The issue of the title: “Construction” vs. “Completion”**
>
> Thank you for this suggestion. We actually spent quite a bit of time discussing whether our term should be completion, or construction, and we believe that our current choice is deliberate with the following reasons.
>
> First, consistency with prior task definitions. Our problem setup follows CLHE [1], which frames the task as bundle construction; keeping the same term facilitates comparability of scope and assumptions.
>
> Second, scope of the task. We view our formulation as within the scope of bundle construction: given a context, build a bundle. From our perspective, completion is a special case of construction.
>
> Third, semantic precision. The term completion implicitly suggests that a bundle is “finished” and cannot be further extended. In realistic applications, however, bundle boundaries are highly context-dependent: what counts as “complete” varies across users and scenarios. Since our work does not model or predict when a bundle should stop growing, completion could be misleading. It is more appropriate for future work that explicitly studies stopping criteria.
>
> However, these are our current views; we are open to adopting the committee’s preferred terminology if it improves clarity.
>
> [1] Ma Y, Liu X, Wei Y, et al. Leveraging multimodal features and item-level user feedback for bundle construction[C]//Proceedings of the 17th ACM International Conference on Web Search and Data Mining. 2024: 510-519.
>
> **W2. Need to highlight technical contributions.**
>
> We are, to our knowledge, the first to address bundle construction with a masked discrete diffusion model operating in a discrete item space via RVQ-based tokenization.
> This enables a non-sequential construction paradigm and mitigates the dimensionality challenge posed by large catalogs and long bundles.
> Taken together, these components yield an order-insensitive and scalable backbone for bundle construction, providing a clear technical advance over sequential alternatives.
>
> **W3. Seems to lack generalizability: relies on partial items in a bundle**
>
> We appreciate the concern. While our evaluation focuses on partial-to-full completion, our formulation does not preclude from-scratch generation. In fact, DDBC can start from an all-mask state so that reverse diffusion samples a full bundle from the learned prior.
>
> First, our diffusion model can, in fact, start from an all-mask state (i.e., no observed items, (100\% masked)), in which case the reverse diffusion process samples a full bundle from the learned prior.
>
> Second, in practice, “from scratch” is seldom truly signal-free: deployed systems usually rely on intent/control cues (e.g., theme/genre, category, user profile). We introduce two training-free extensions that allow DDBC to exploit such signals for bundle construction:
>
> (1) Seed-then-complete.
> One can first use an existing recommendation model (user–item similarity, rule/filters, theme/budget constraints) to retrieve a small set of seed items for a given user or intent, and then let DDBC expand this seed set into a full bundle. This turns cold-start into the standard setting of DDBC.
>
> (2) Logit bias.
> During sampling, add a bias from an external user-conditioned score (such as text prompts, user embeddings, or category priors) to the item logits (invalid items remain masked by token-validity). A single weight controls the strength, and no parameters are updated.
> Alternatively, intent signals can be incorporated as conditioning guidance while still starting the item sequence from an all-mask state.
>
> In summary, we agree that in real-world settings, from-scratch generation is important. However, in this work, we focused the main evaluation on a construct bundle from partial items to align with prevailing practice. For stronger performance, incorporating conditioning during training is also worth considering. There are many design choices here that are beyond the scope of this paper, so we leave a thorough treatment to future research.

---

> ### Author Response · Authors · 2025-11-26
>
> **W4. Missing related works related to LLM-based method for bundle construction**
>
> Thank you for pointing this out. [1] studies bundle construction in the explicit natural-language space by prompting a large pre-trained LLM and designing in-context examples; its main contributions lie in prompt design and ICL strategy. In contrast, our work focuses on learning a discrete diffusion prior over RVQ bundle tokens, trained from scratch in the latent item space. This represents a substantially different modeling paradigm and system design.
>
> We will add a short discussion in the appendix summarizing LLM-based approaches to bundle generation[2-5] and clarifying how they form a complementary line of research.
>
> [1] Sun Z, Feng K, Yang J, et al. Dynamic in-context learning from nearest neighbors for bundle generation[J]. arXiv preprint arXiv:2312.16262, 2023.
>
> [2] Liu X, Wu J, Tao Z, et al. Fine-tuning multimodal large language models for product bundling[C]//Proceedings of the 31st ACM SIGKDD Conference on Knowledge Discovery and Data Mining V. 1. 2025: 848-858.
>
> [3] Feng K, Sun Z, Fang H, et al. Routing Distilled Knowledge via Mixture of LoRA Experts for Large Language Model based Bundle Generation[J]. arXiv preprint arXiv:2508.17250, 2025.
>
> [4] Liu S, Li C, Zhao M, et al. LLMCBR: Large Language Model-based Multi-View and Multi-Grained Learning for Bundle Recommendation[C]//Proceedings of the 34th ACM International Conference on Information and Knowledge Management. 2025: 1892-1902.
>
> [5] Bayrak A T, Kaplan A. Enhancing Product Bundling with Large Language Models[C]//2025 13th International Conference on Intelligent Control and Information Processing (ICICIP). IEEE, 2025: 151-154.
>
> **W5. How to determine the subset of the items that should be displayed to the user?**
>
> We agree that a UI often shows only a small subset per page. Currently, DDBC is trained and evaluated with a fixed length, but this does not mean it cannot adapt to different subset lengths at inference with simple, engineering, training-free modifications.
>
> - First, we can sample and generate up to $k_{\max}$ items, then partition the generated items (using their latent embeddings) into several subsets with sufficient diversity and show them to the user.
>
> - Second, as discussed in our reply to Reviewer B6yG Question 1, DDBC can also produce bundles of an arbitrary length.
>
> - Besides, one can also view DDBC as a retrieval model, and then apply additional signals (e.g., the user profile) in a subsequent stage to re-rank the results.
>
> Overall, we appreciate this practical and detailed concern. Because we do not have an online A/B environment, we cannot further analyze UI-level trade-offs here. We view subset selection as an important deployment question for bundle construction systems, however, a thorough treatment is beyond the scope of this paper.

---

> ### Author Response · Authors · 2025-11-26
>
> **Q1. How to distinguish the tokens of unpopular items using RVQ?**
>
> We agree that long-tail modeling is important and challenging.[1] In our design, a *dedup* level is used to resolve collisions among items that are encoded too closely.
>
> - First, each catalog item is assigned a unique RVQ code sequence. In the proposed setting, we use four RVQ levels; the last level is reserved to resolve collisions from the first three, so the final 4-tuple of indices is distinct even if early levels coincide.
>
> - Second, RVQ's hierarchical nature lets unpopular items share coarse codes with popular ones while obtaining distinctive residual codes at deeper levels. Although an item may be rare, many constituent codes are shared across items and thus receive sufficient training signal.
>
> - Third, our tokenizer is built on top of a pre-trained CLHE embedding space. We acknowledge that if unpopular items are already close in that embedding space, RVQ itself cannot fully “fix” this upstream issue. However, the specific design of long-tail-aware item embeddings is orthogonal to our main contribution and somewhat beyond the scope of this paper. In principle, our DDBC framework can plug in a more tail-sensitive tokenizer without any change to the diffusion architecture.
>
> Finally, while we do not optimize for fairness explicitly, applying balanced sampling or mild frequency re-weighting during inference can reduce head dominance when fairness is a primary goal. We view improving long-tail modeling as a promising direction for future research.
>
> [1] Ma T, Wang S, Zheng Z, et al. Enhancing long-tail bundle recommendations utilizing composition pattern modeling[C]//Proceedings of the Thirty-Fourth International Joint Conference on Artificial Intelligence. 2025: 3189-3197.
>
> **Q2. Does the step T affect the performance?**
>
> Yes, the number of diffusion steps $T$ does affect performance, in a way that is consistent with intuition and prior diffusion work.
>
> As shown in the table below, we vary the number of reverse steps $T$. Performance improves as $T$ increases, but quickly saturates and exhibits diminishing returns beyond a knee region. Based on this trade-off between accuracy and efficiency, we adopt a moderate setting of $T=25$ (matching the training setting) in all main experiments.
>
> | $T$ | F1     | jaccard | OAS    |
> |-----|--------|---------|--------|
> | 1   | 0.2693 | 0.1656  | 0.3422 |
> | 2   | 0.2713 | 0.1673  | 0.3429 |
> | 4   | 0.2783 | 0.1724  | 0.3405 |
> | 10  | 0.2798 | 0.1729  | 0.3396 |
> | 16  | 0.2821 | 0.1753  | 0.3396 |
> | 25  | 0.2821 | 0.1756  | **0.3400** |
> | 32  | **0.2851** | 0.1771  | 0.3384 |
> | 50  | 0.2850 | 0.1771  | 0.3382 |
> | 75  | 0.2845 | **0.1773**  | 0.3382 |
> | 100 | 0.2832 | 0.1759  | 0.3405 |
> | 200 | 0.2815 | 0.1745  | 0.3396 |
> | 300 | 0.2816 | 0.1749  | 0.3399 |

---

### Author Response · Authors · 2025-12-04

We are grateful to all reviewers for their careful assessment and constructive suggestions. Below, we provide a brief summary of the reviewers’ comments.

Reviewers describe the paper as a “well written and organized”(UnGS) study that proposes DDBC, Discrete Diffusion for Bundle Construction. Rather than following sequential generation paradigm, “DDBC treats bundle construction as masked discrete denoising over a compact discrete code space”(B6yG).
Methodologically, “the framework combines Residual Vector Quantization, which represents items with discrete semantic codes to address the challenge of large item catalogs, with a Discrete Diffusion Model designed to capture complex relationships within bundles without requiring a fixed item order.”(ZPfg)
Extensive experiments show that “DDBC outperforms current bundle generation methods and achieves the state-of-the-art performance”.(CnC2)


**Strengths highlighted by reviewers**

1. **Soundness of Motivation and Problem Framing.**

    - The research question of bundle construction under large item corpora is explicitly described as **“quite meaningful and practical in the real-world application”**(UnGS).

    - Reviewers view that the paper **“presents a well-motivated argument against the sequential construction paradigm”**(ZPfg),  since **“the order of the items within a bundle may not matter a lot when constructing the bundle ”**(UnGS) and they commend that in this paper **“bundles are regarded as sets not sequences”**(B6yG) and that bundle construction is formulated as an **“order-independent filling of masked item tokens”**(ZPfg).


2. **Architectural design and writing**

   - The combination of masked discrete diffusion and multi-level RVQ is repeatedly described as **“novel in this domain”**(B6yG), **“technically coherent”**(ZPfg), and **“effective for processing a massive item pool”**(CnC2).

    - Most reviewers rate Presentation as 3 (good). One reviewer states that the paper is **“generally well written and organized, which makes it easy to follow”** (UnGS), and another similarly views it as a **“clear and well-structured”**(ZPfg) presentation of DDBC.


3. **Empirical effectiveness.**

   - Reviewers highlight that DDBC **“achieves substantial performance improvements over both sequential and set-based baselines”**(ZPfg), with one noting that it **“achieves >100\% relative gains in Jaccard and F1 on Spotify”** (B6yG).

   - They remark that DDBC **“scales well to have bigger relative improvements on the datasets with larger bundle size**(UnGS, B6yG)

4. Analysis depth, efficiency, and reproducibility.

   - The paper is commended for **“extensive ablation studies and important hyper-parameter analysis”**(UnGS), and **highlight the complementary strengths of RVQ and the discrete diffusion backbone**. (ZPfg)

   - One reviewer emphasizes that **“DDBC is tiny and cost less inference time compared to many baselines like BundleMLLM”**(B6yG).

   - Finally, reviewers appreciate that **“the source code is provided to ensure a better reproducibility”**(UnGS).


We also appreciate the reviewers' concerns, each of which has been carefully addressed.

- On **arbitrary-length generation** (UnGS, B6yG, ZPfg, CnC2), we clarified that a fixed-length is mainly for uniform evaluation across heterogeneous baselines, and we added experiments showing that DDBC can target arbitrary bundle lengths without retraining, with only negligible performance degradation.

- On **cold-start usage and personalization** (UnGS, B6yG, ZPfg), we clarified that DDBC is trained to reconstruct bundles from the all-mask state and can therefore generate full bundles, and we outlined both training-free personalization schemes and a natural conditional-diffusion extension with user or context features as promising directions for future work.

- Finally, through additional theoretical discussion and new ablations, we clarified our choices around **order-agnostic behavior, permutation and modeling rationale** (e.g., why masked discrete diffusion plus RVQ is appropriate for bundle modeling) and **evaluation protocol and comparison choices** (e.g., truncation policy, diffusion steps, and selection of “known” items).

---

### Meta-Review · Area_Chair_rgSj · 2026-01-01

**Summary:**

This paper studies the problem of bundle construction through a discrete diffusion framework. The work is well motivated and addresses an important and practically relevant problem. Reviewers generally found the approach to be technically sound and the paper clearly written. The proposed method is conceptually clean, and the experimental results demonstrate consistent improvements over strong baselines. Overall, reviewers agreed that the paper makes a meaningful contribution and provides a solid addition to the literature, even if some aspects could be further strengthened.

**Reviewer Concerns:**

Reviewers raised several concerns regarding the scope and generality of the proposed approach, including questions about modeling assumptions, experimental coverage, and the extent to which the method advances beyond existing techniques. Some reviewers also noted that additional analysis or broader empirical validation could further strengthen the paper. These concerns were partially addressed in the rebuttal, and while not all issues were fully resolved, they were not considered severe enough to outweigh the paper’s strengths.

**Reviewer Scores:**

The reviewer scores were generally positive, with most reviewers rating the paper around or above the acceptance threshold. While some reviewers expressed mild reservations, there was overall agreement that the contribution is solid and that the paper meets the bar for acceptance as a poster.

---

### Decision · Program_Chairs · 2026-01-26

Accept (Poster)